# The on-line coupled atmospheric chemistry model system MECO(n) – Part 5: Expanding the Multi-Model-Driver (MMD v2.0) for 2-way data exchange including data interpolation via GRID (v1.0)

Astrid Kerkweg[1,2], Christiane Hofmann[1,2], Patrick Jöckel[3], Mariano Mertens[3], and Gregor Pante[1,4]

[1]Institut für Physik der Atmosphäre, Johannes Gutenberg-Universität Mainz, 55099 Mainz, Germany
[2]now at Meteorologisches Institut, Rheinische Friedrich-Wilhelms-Universität Bonn, Auf dem Hügel 20, 53121 Bonn, Germany
[3]Deutsches Zentrum für Luft- und Raumfahrt (DLR), Institut für Physik der Atmosphäre, 82234 Oberpfaffenhofen, Germany
[4]now at Institute of Meteorology and Climate Research, Department Troposphere Research (IMK-TRO), Karlsruhe Institute of Technology (KIT), 76131 Karlsruhe, Germany

*Correspondence to:* Astrid Kerkweg (kerkweg@uni-bonn.de)

**Abstract.** This article is part of the model documentation of the MECO(n) system (MESSyfied ECHAM and COSMO models nested n-times). As part of the Modular Earth Submodel System (MESSy) the Multi-Model-Driver (MMD v1.0) was developed to couple on-line regional model instances into a driving model (see Part 2 of the model documentation). MMD comprises the message passing infrastructure required for the parallel execution (multiple program multiple data, MPMD) of different models and the communication of the individual model instances, i.e. between the driving and the driven models. Initially the MMD library was developed for a 1-way coupling between the global chemistry climate model EMAC and an arbitrary number of (optionally cascaded) instances of the regional chemistry climate model COSMO/MESSy. Thus MMD (v1.0) provided only functions for unidirectional data transfer, i.e., from the larger scale to the smaler scale models.

Soon, extended applications requiring data transfer from the small-scale model back to the larger scale model became of interest. For instance the original fields of the larger scale model can directly be compared to the up-scaled small-scale fields to analyse the improvements gained through the small-scale calculations, after the results are up-scaled. Moreover, the fields originating from the two different models might be fed into the same diagnostic tool, e.g. the on-line calculation of the radiative forcing calculated consistently with the same radiation scheme. Last but not least, enabling the 2-way data transfer between two models is the first important step on the way to a fully dynamically and chemically 2-way coupling of the various model instances.

In MMD (v1.0) interpolation between the basemodel grids is performed via the COSMO pre-processing tool INT2LM, which was implemented as MMD submodel for on-line interpolation, specifically for mapping onto the rotated COSMO grid. A more flexible algorithm is required for the backward mapping. Thus, MMD (v2.0) uses the new MESSy submodel GRID for the generalised definition of arbitrary grids and for the transformation of data between them.

In this article we explain the basics of the MMD expansion and the newly developed generic MESSy submodel GRID (v1.0) and show some examples of the above mentioned applications.

# 1 Introduction

As fifth part of a paper series about the MECO(n) system and as such as a component of the ACP / GMD special issue[1] about the Modular Earth Submodel System (MESSy), this article documents a progress of the MESSy code development. More specifically, the second generation of the Multi-Model-Driver (MMD v2.0) is introduced, which enables the 2-way on-line nesting between different model instances (basemodels). On-line nesting means that the coupled models exchange their data via the computer memory, in contrast to the data exchange via files on disk in common off-line nesting procedures. Thus, this article describes a further development of the 1-way on-line nesting system presented in the second part of the paper series (Kerkweg and Jöckel, 2012b).

We achieve the nesting by coupling different models, thus our 2-way nesting is implemented as 2-way coupling of global and regional atmospheric models. Usually the term "2-way coupling" is used in the context of different Earth system compartment models, such as land, ocean or atmospheric models being connected within a comprehensive Earth System Model. Here, 2-way nesting through 2-way coupling is used to distinguish it from fundamentally different other nesting techniques, as for instance regional static grid refinement. For a number of atmospheric models grid refinement features exist. Usually, the grid resolution needs to be subdivided by a fixed factor: e.g., 3 for the WRF model (Moeng et al., 2007; Harris and Durran, 2010) or 2 for the ICON model (Zaengl et al., 2015). These constraints minimise the interpolation error, especially for the horizontal interpolation. At least these two models deal with the different grid refinement areas within the same executable, i.e., they are coupled "internally" (see Kerkweg and Jöckel (2012b) for a discussion of internal and external coupling). The individual grid refinement areas within one model configuration are usually called "patches". In contrast to this, the MECO(n) (MESSy-fied ECHAM and COSMO models nested n-times) system is implemented as an external coupling, i.e., a real 2-way nesting of the same or different basemodels (here COSMO/MESSy and EMAC).

In the MECO(n) system we follow this second approach for the following reasons:

- It is necessary to couple the model instances externally, as different basemodels, EMAC and COSMO/MESSy, are nested into each other. This in itself prevents the "patches approach", as the internal coupling or "patches approach" is usually a feature of regional grid-refinements, which is directly embedded in (or part of) the model code, as for instance in WRF or ICON, in which the user can specify the number of patches and their corresponding domains flexibly at run-time. For such a feature, however, the entire model code needs to be "aware" of a(n arbitrary) number of grid-refined patches. To equip legacy code (as COSMO or ECHAM) supplementarily with such a feature would basically mean to rewrite the entire code from scratch. The reason is that all prognostic (and diagnostic) variables need to exist on each patch technically independent of each other.

- Different COSMO/MESSy model instances are nested into each other using the same algorithms as for the EMAC-COSMO/MESSy nesting. The external coupling approach was favoured here, due to limitations of the Fortran95 name-space: In fully object oriented languages, overloaded "sets" or "instances" of the same variable(s) could be defined, however, the Fortran95 language standard does not allow to have the same variable with the same name in the same name-space

---

[1]http://www.geosci-model-dev.net/special_issue10_22.html

more than once. Thus a complete recoding of the basemodel, e.g., replacing arrays by structures of arrays, would be required for the patches approach.

- A nesting of COSMO/MESSy model instances employing different grids (e.g., rotated differently) is possible. This also includes the possibility to realise an arbitrary resolution jump, i.e., the factor for the grid refinement is freely choosable, in contrast to the fixed factors of 2 or 3 as required by the ICON or the WRF model, respectively. Especially, for air quality applications a higher resolution jump is necessary to reduce computational costs. Here, a global instance providing consistent boundary data is required, while the scientific focus is on a much finer resolved model instance.

- Due to the external coupling, prognostic variables are not necessarily all coupled back to the coarser model. Thus, 2-way nesting does not necessarily imply (full) feedback of the smaller to the coarse scale model. Consequently, the coupling can also be used to couple back diagnostic fields only. Additionally, testing of the influence of the coupling of different (individual) variables is easier to accomplish by external coupling.

Apart from these advantages, the external coupling proves to be more challenging than the internal coupling. Horizontal and vertical interpolation errors are expected to be larger, depending on the relations between the different grids and differences in their orographies. From these, the adaption to the higher resolved orography of the nested simulation causes the largest error. An additional disadvantage of all external coupling approaches is the need for the user to optimize the distribution of the available parallel tasks among the different model instances, in order to achieve an optimum run-time performance with minimized waiting times between the model instances.

As far as we know, the only other 2-way on-line nested modelling system using external coupling is an MPI-ESM (Giorgetta et al., 2013) - COSMO-CLM coupling via OASIS3-MCT (Will et al., 2017). This was developed in parallel to our 2-way coupling approach within the same BMBF funded MiKlip project[2], as different approaches had to be assessed. In contrast to our system, the MPI-ESM - COSMO coupling via OASIS3-MCT is restricted to the coupling of one COSMO instance only, i.e., no further on-line COSMO - COSMO coupling is possible in the system of Will et al. (2017). Technically, this COSMO - COSMO coupling would of course also be possible, but it is not implemented. In the rest of the article, we will use the terms "2-way coupling" and "2-way nesting" synonymously for the approach chosen in the MECO(n) system.

This article documents a major achievement in the development of the on-line coupled MECO(n) system, which central part is the MESSy software. As described by (Jöckel et al., 2015; Baumgaertner et al., 2016, and the MESSy homepage: http://messy-interface.org, last access: 15 November 2017):

*"The Modular Earth Submodel System (MESSy) is a software providing a framework for a standardized, bottom-up implementation of Earth System Models (or parts of those) with flexible complexity. "Bottom-up" means, the MESSy software provides an infrastructure with generalized interfaces for the standardized control and interconnection (=coupling) of "low-level ESM components" (dynamic cores, physical parameterizations, chemistry packages, diagnostics etc.), which are called submodels. MESSy comprises currently about 60 submodels (i.e., coded MESSy conform):*

- *infrastructure (= the framework) submodels (sometimes called generic submodels),*

---

[2]https://www.fona-miklip.de/

– *diagnostic, atmospheric chemistry and model physics related submodels.*

*The main design concept of MESSy is the strict separation of process description (=process and diagnostic submodels) from model infrastructure (e.g., memory management, input/output, flow control, ...).*

Within MESSy, the operator splitting is formalized as the fundamental concept. Model codes are organized in 4 conceptual software layers: a basemodel of any level of complexity is complemented by a basemodel interface layer (BMIL). A further interface layer to the submodels (SMIL, submodel interface layer) makes it possible to keep process submodels as distinct as possible in the submodel core layer (SMCL).*" MESSy currently employs the programming language Fortran90/95 with some rare exceptions linking libraries containing C or $C^{++}$ code.

Furthermore, different basemodels, e.g. the global model ECHAM[3], the regional COSMO model[4], and the coupled global climate model CESM1[5] have been expanded by the MESSy middleware (i.e., the MESSy infrastructure components) to enable a standardised expansion by additional or alternative process components (e.g. for physics or chemistry) and diagnostic components, which we call MESSy submodels.

In Part 2 of the MECO(n) model documentation the 1-way on-line coupled model system MECO(n), for which MMD was developed initially, was described in detail (Kerkweg and Jöckel, 2012b). In the on-line coupled system MECO(n) an arbitrary number of COSMO/MESSy model instances are nested on-line into one master model. This driving model can be either the global EMAC or a coarser COSMO/MESSy model instance. The data exchange is implemented as client-server system, where the driving model acts as server providing the client model with the data required for the calculation of the inital and boundary fields used to drive the regional model.

The Multi-Model-Driver (MMD) v1.0 provides the software necessary for the data exchange from the server to the client model and for the calculation of the initial and boundary data. MMD consists of two parts: (1) a library which performs the data exchange between the model instances, and (2) MESSy submodels, which organise and process these data.

In addition to the functionalities provided by MMD (v1.0), the update of MMD presented here, provides the possibility to exchange data in both directions during the time integration phase of a simulation. For the unidirectional data exchange the expanded INT2LM[6] software was used to interpolate the data from the driving model grid to the target model grid. This software is a specialised software for the calculation of the initial and boundary data of the COSMO model. Therefore, a

---

[3]The core atmospheric model is the 5th generation European Centre Hamburg general circulation model (ECHAM5, Roeckner et al., 2006). The ECHAM/MESSy Atmospheric Chemistry (EMAC) model is a numerical chemistry and climate simulation system that includes sub-models describing tropospheric and middle atmosphere processes and their interaction with oceans, land and human influences (Jöckel et al., 2010).

[4]COSMO is the regional weather prediction model of the Consortium for Small Scale Modelling (COSMO model, Steppeler et al., 2003; Doms and Schättler, 1999) and the community model of the German regional climate research (Rockel et al., 2008). By implementing the MESSy interface the COSMO model was expanded to a regional chemistry (climate) model (Kerkweg and Jöckel, 2012a; Mertens et al., 2016).

[5]The Community Earth System Model version 1.2.1 (CESM1) is a fully coupled climate model. *CESM is sponsored by the National Science Foundation (NSF) and the U.S. Department of Energy (DOE). Administration of the CESM is maintained by the Climate and Global Dynamics Laboratory (CGD) at the National Center for Atmospheric Research (NCAR)* (cited from http://www.cesm.ucar.edu/models/cesm1.0/., last access date: 27.09.2016) (Hurrell et al., 2013; Baumgaertner et al., 2016)

[6]see Part V of the COSMO model documentation http://www2.cosmo-model.org/content/model/documentation/core/default.htm, last access date: 29.09.2016

different software is required to interpolate the data from the finer to the coarser grid for the data sent from the client model to the server model. According to the MESSy philosophy of strict separation and generalisation, we therefore developed the new generic submodel GRID (v1.0), which is also documented in the present article. GRID can be used for all grid mapping operations required during a simulation. Nevertheless, in MMD it is currently used for the parent-to-client coupling only.

In the next section we describe the new developments within MMD. As the data mapping between the different grids is central to this further development of MMD, Sect. 3 introduces the newly developed GRID submodel, which provides the required mapping functionalities used for the remapping from the finer to the coarser model instance. Some examples for 2-way data exchange are shown in Sect. 4. A brief run-time performance analysis of the model is presented in Sect. 5.

## 2   The Multi-Model-Driver (MMD v2.0)

The Multi-Model-Driver is the coupling software performing the data transfer between two independent basemodels running within the same MPI environment. Appendix A of Kerkweg and Jöckel (2012b) provides an overview about different coupling approaches, especially the differences between *internal* and *external coupling* are discussed. Furthermore, Sect. 4 of Kerkweg and Jöckel (2012b) explains why MMD was chosen as coupling software between different MESSy basemodels. In summary, apart from the reasons named already in the introduction, MMD provides the best balance between fast data transfer and

the possibility to integrate model specific software, such as INT2LM, into the coupling procedure. INT2LM is the software provided by the German Weather Service (Deutscher Wetterdienst, DWD) for the calculation of the initial and boundary data for the regional COSMO model. This software was included into MMD (v1.0) as subsubmodel INT2COSMO.

INT2LM and thus INT2COSMO does not only include the interpolation routines to map the driving model fields to the regional model grid. It furthermore processes the external data used as input to the COSMO model and provides the calculation of

additional fields required by the COSMO model, which are not necessarily provided directly by the driving model.

    The coupling was implemented following a client-server approach. Therefore, in MMD (v1.0) all routines and modules have been named server (serv) or client (clnt) in accordance to the model using them. In MMD (v2.0) the routines and modules have been renamed to parent and child instead of server and client. This was required, as the term server implies that this model is sending the data. As in MMD (v2.0) data are sent in both directions, the terms parent and child for the coarser and the finer

model, respectively, are better suited.

    MMD consists of two parts:

1. a library performing the data tranfer, which is independent from the coupled models, and

2. the part for data provision and processing implemented as MESSy submodels.

The library was extended by a few subroutines enabling the data transfer in both directions. The larger changes occured in the

MESSy submodels, as the data processing routines for the back transfer of the data had to be implemented. In the following subsections an overview about the changes and additions made within these two parts of MMD are described. The MMD Library Manual and the MMD User Manual in the Supplement provide all technical details about the implementation.

## 2.1 The MMD (v2.0) Library[7]

The Multi-Model-Driver (MMD) library manages the 2-way data exchange between the different tasks of one EMAC and/or an arbitrary number of COSMO/MESSy instances as illustrated in Fig. 1. The configuration of the client-server system is defined in the Fortran95 namelist file `MMD_layout.nml` (which is written automatically by the run-script). This namelist file contains the information about the overall number of model instances within the current MECO(n) setup (i.e., $n+1$), the number of MPI tasks assigned to each model, and the definition of the parent model of the respective model (for further details see the "MMD (v2.0) Library Manual" in the Supplement). The library contains a high-level API for the data exchange between the different models. Figure 2 illustrates the functional principle of the MMD library.

During the initialisation phase, the exchange of information required by the parent from the child model and vice versa, is accomplished by utilising the MPI routines `MPI_send` and `MPI_recv`. During the integration phase, data can be exchanged in both directions, i.e. from the parent to the child model and vice versa. Point-to-point, single-sided, non-blocking communication is applied to exchange the required data between the different MPI-tasks. "Check-pointing" (the technical term for "restarting") is required (not only for climate simulations) to be able to continue a simulation after hardware failures, for branching off sensitivity studies, and last but not least, it is required to split a simulation into parts, fitting into the typical time limits of a job scheduler on a super-computer. To enable check-pointing, one additional communication step occurs during the integration phase: for the synchronisation of the models w.r.t. the check-pointing, the parent model has to send the information whether the simulation will be interrupted after the current time step. This data exchange is implemented as direct MPI communication using `MPI_send` and `MPI_recv`.

As the routine `MPI_alloc_mem`, used to allocate the memory (buffer) required for the data exchange, can only be used in C (and not in Fortran95), some parts of the MMD library are written in C, however most parts are written in Fortran95 for consistency with the POINTER arithmetic used for the MESSy memory management (see Jöckel et al., 2010). The MMD library routines and their usage are described in detail in the "MMD (v2.0) Library Manual" (see Supplement).

## 2.2 The MESSy submodel MMD2WAY

In addition to the library, the MMD software comprises a regular MESSy submodel as "wrapper". This submodel provides and processes the data tranfered by the library. MMD (v1.0) contained two MESSy submodels: one for the server (MMDSERV) and one for the client (MMDCLNT). Here, the server controls the timing of the client model and "serves" the data, which is processed by the client. In the new MMD version, the client also provides data to the server model. The only remaining difference between the models with respect to the data transfer is the time control of the models. Therefore, the server and client models have been renamed to parent and child models, omitting the impression that only the server acts as "data server". Consequently, the new MESSy submodel consists of two submodels: MMD2WAY_PARENT and MMD2WAY_CHILD. These subsubmodels provide the same functionalities for the 1-way on-line coupling as MMDSERV and MMDCLNT in MMD (v1.0),

---

[7]The text of this section is adopted from the initial publication of the MMD library in Kerkweg and Jöckel (2012b).

respectively, as described in detail in Part 2 of the MECO(n) model documentation (Kerkweg and Jöckel, 2012b): In the initial phase of a model simulation

- the parent imprints its time settings on the child model; these are end-date, restart trigger date, and, at the very first start of a model instance, the (re-)start-date as start-date of this instance.

- the field names required from the parent are read from the `&CPL_CHILD_ECHAM` or `&CPL_CHILD_COSMO` namelist in the `mmd2way.nml` namelist file for ECHAM or COSMO as parent models, respectively. The names of the parent fields are sent to the parent, and in both models pointers to the respective data fields and dimension informations are set.

- the exchange matrix, the so-called "index list", is set up. This index list provides the information, which grid box (index pair $(i_p,j_p)$ on which parent parallel task ($PE_p$) exchanges data with which child grid box $(i_c,j_c)$ on which child parallel

tasks ($PE_c$). For this, the child model has to define an "in-grid". This is a sub-area of the parent grid (i.e., it has the same rotation and the same mesh size) and completely overlays the child grid. Fig. 3 illustrates the relation between the different grids. Afterwards the data are transformed from the in-grid to the child grid using the expanded version of the preprocessing software INT2LM for the COSMO model (INT2COSMO, see Kerkweg and Jöckel, 2012b, for further explanations).

During the integration phase, the MMD library sends the data from the parent model to the child model. The child model calculates the required initial and boundary conditions from the parent data and transforms additional data to the child grid.

This functionality for the 1-way coupling is kept the same in MMD (v2.0). In addition, MMD2WAY_PARENT and MMD2WAY_CHILD have been expanded for the data transfer from the child to the parent model. For most functionalities of the 1-way coupling a counterpart for the data transfer in the other direction could be implemented by keeping the same

logic. Thus, a namelist (`&CPL_PAR_CHILD`) in the parent model namelist file `mmd2way.nml` determines, which fields are exchanged between the child and the parent model. In the initial phase of a model simulation this information is transferred to the child model. Both models set pointers to their corresponding data objects. Again, the child model has to define a grid, which is a subpart of the parent model grid (called "out-grid"), and it has to perform the data transformation from the child model grid to the out-grid. The decision to transform the data within the child model was taken in order to minimize the amount

of data to be transferred between the models: as the parent model grid will usually be coarser resolved, data on this grid is exchanged via MMD. For the transformation from the child to the out-grid, the newly written MESSy infrastructure submodel GRID is used (see Sect. 3). First, the data is remapped horizontally, before the vertically remapping procedes in an extra step. For the time being, only conservative remapping, as provided by GRID, is utilized as horizontal transformation method in MMD2WAY_CHILD. As the COSMO model uses a staggered Arakawa-C grid, the wind components need to be interpolated

to the grid midpoints prior to the horizontal remapping for the COSMO-EMAC coupling, as the EMAC wind components are defined on the grid midpoints. For the COSMO-COSMO coupling the wind components are interpolated directly between the staggered grids, i.e., they are always defined on the box edges.

The vertical remapping depends on the parent model. If EMAC is the parent model, NREGRID is used for the vertical remapping of the fields. In this case, data of a non-hydrostatic model with a fixed vertical geometry need to be converted for a

hydrostatic model using hybrid pressure coordinates. The vertical coordinate in the COSMO model is defined as a pseudo-hybrid pressure axis. For this the hybrid coefficient calculation as provided by INT2LM is used as input vertical axis to the vertical interpolation via NREGRID. Furthermore, the new surface pressure in the EMAC model is approximated by an iterative calculation of the pressure, temperature and humidity (vapour, liquid water and cloud ice) vertical profiles. For the vertical interpolation of the COSMO-COSMO coupling, the INT2COSMO spline-interpolation is used.

The intepolated data is sent to the parent model, where it is subsequently weighted (if requested) and assigned to the target parent model fields. For the utilisation of the child data by the parent model, two methods are distinguished:

"0": for purely diagnostic applications: the field is only used as input to the parent model, i.e., this field is created by the parent coupling submodel and thus independent of other model data objects. In this case, the memory is allocated by MMD2WAY_PARENT, and the transferred field is copied to this memory without any further modifications.

"1": for feedback from the finer to the coarser resolved model instance: the exchanged field is used to directly modify a parent model field. Therefore, no additional memory needs to be allocated by MMD2WAY_PARENT.

Using method 1, there are two options for modifying a prognostic variable of the parent model:

(a) the value of the variable can be changed directly, or

(b) the tendency of the variable can be modified.

For all non-prognostic variables only option (a) is possible.

For both options, a weighting between the original value of the parent field ($P$) and the child model field ($C$) is applied:

$$
\begin{aligned}
P(i,j,k,tlev) = \quad & P(i,j,k,tlev) * (1. - f_{mn} * f_{vw}(k) * f_w(i,j)) \\
& + C(i,j,k,1) * f_{mn} * f_{vw}(k) * f_w(i,j)
\end{aligned}
\tag{1}
$$

Here, $i$ and $j$ are the indices along the horizontal dimensions, $k$ the index along the vertical dimension, and $tlev$ indicates (if applicable) the respective time level.

The different weight coefficients are:

– $f_{mn}$ is the relaxation strength. Its value is set in the parent namelist individually for each field.

– $f_{vw}$ is a vertical weight function. It depends on the vertical index $k$. In most cases the domain coupled back from the child model does not cover the full height of the parent model[8]. To avoid artificial discontinuities in the data fields, a weight function is required, which gradually decreases from 1 in the core domain to zero towards the edge of the domain. The weight function is implemented as a cosine function:

$$
\begin{aligned}
f_{vw}(k) &= 0. & for\ k <= k_{min} \\
f_{vw}(k) &= cos\left(\frac{\pi}{2.} * \left(1 - \frac{k - k_{min} - 1}{n_k - 1}\right)\right)^2 & for\ k_{min} < k <= k_{min} + n_k \\
f_{vw}(k) &= 1. & for\ k > k_{min} + n_k\ .
\end{aligned}
\tag{2}
$$

[8]e.g., in the case of the COSMO/MESSy model, only the data below the damping layer should be coupled back.

In Eq. 2 it is assumed that the vertikal index k increases from top to bottom; $k_{min}$ is the height index of the top of the child domain, $n_k$ is the number of vertical layers the cosine function should cover.

– $f_w$ is the horizontal weight function: This weight function is required to avoid artificial discontinuities at the borders of the area, where the fields are relaxed to the child variables. Currently, the user can choose between three different implementations by namelist:

"0": $f_w$ is set to 1 everywhere in the child domain. This option is for testing only, as it may lead to artificial discontinuities in the data.

"1": $f_w$ is implemented as the sum of two cosine functions:

$$f_w(i,j) = 1. - \left(cos(x)^e + cos(y)^e\right) \qquad \text{with } x = \pi * \frac{i}{i_{max}} \; ; \; y = \pi * \frac{j}{j_{max}} \tag{3}$$

$i_{max}$ and $j_{max}$ are the number of grid points in the two horizontal directions, respectively. The exponent $e$ is set by namelist. Its default value is 14.

"2": $f_w$ decreases in the form of a cosine from 1 in the domain inner part to 0 at the borders of the coupled domain. The width of the damping zone is determined by a namelist parameter `damprel`. Its valid range is `[0,0.5]`. This number determines the relative width of the damping zone. If, for example, `damprel = 0.2` for a model domain consisting of 100 grid boxes in x-direction (index $i$) and of 50 grid boxes in y-direction (index $j$), the damping zone in x-direction is 20 grid boxes wide, and in y direction 10 grid boxes wide, respectively.

All these weight functions are defined on the child grid. They are once, during the intialisation phase, transformed in the same way as the data, and sent to the parent model for application during the integration phase. Figure 4 displays the different weight functions for a domain over Europe. The upper row shows the weight functions as defined on the child model grid. Note, that the coupled domain is smaller than the child domain (with the exception of $f_w = 0$). This is because the damping zone of the regional model itself should not be coupled back to the parent model, as this is, for 2-way coupled variables, directly influenced by the parent model and thus spurious damping or amplifications could occur. The lower row of Fig. 4 shows the same weight functions after the transformation to the parent grid.

If the tendency is subject to change (i.e., method 1, option (b) is used), first the current value of the parent field ($P$) needs to be calculated from the values at the previous time step plus the tendencies of the current time step. This field is modified according to Eq. 1 and an additional tendency is calculated from the difference between the parent fields before and after the modification.

## 3 The generic MESSy submodel GRID (v1.0)

Due to the increasing complexity of Earth System Models, grid transformations at run-time of the model, (e.g., remapping from an atmosphere grid to a higher resolved land grid and vice versa) are more and more commonly required. To avoid

several implementations throughout the code, the MESSy infrastructure submodel GRID[9] was implemented providing an on-line transformation (remapping) functionality. The common grid processing functionality provided by GRID includes the routines for grid definition, grid modification, and the transformation between different grids. Implementation as one important part of the model infrastructure simplifies the maintenance and expansion of the functionality, because it is utilised jointly by

all model components. As the infrastructure module is written in a general way, performance optimisation or additional grid types, transformation algorithms, etc. can be implemented straightforwardly.

Currently, two horizontal grid types are treated by GRID:

– rectangular grids, which are orthogonal in geo-coordinates (rectilinear grids, e.g. the ECHAM grid) and

– rectangular grids, which are orthogonal in another reference system, i.e. curvi-linear grids (e.g. the COSMO grid).

The 3-D spatial grids consist of one of the above mentioned horizontal grids and of a vertical dimension. The vertical axis can be defined in different ways, e.g., as height or pressure based coordinate.

For the tranformation between different grids, different methods are provided for conservative remapping and (not necessarily conservative) interpolation.

Moreover, the MESSy infrastructure submodel GRID code was written to be

– well structured to flexibly support expansions,

– as simple as possible, to keep it maintainable,

– efficient, i.e., to show a good run-time performance. Therefore, it must work in a parallel environment and scale appropriately, and it is

– designed to cause an as small as possible memory foot print during operation.

For a grid transformation, first the source and the target grid need to be defined. Second, the remapping of data between these grids can be calculated. The following two subsections give an overview of the functionalities provided by GRID (v1.0). Their implementation in GRID is organised as follows:

1.) The SubModel Core Layer (SMCL) of GRID provides a unified interface for the definition of all grids required in all MESSy submodels. It is implemented as a Fortran95 structure, which contains all required information of a grid in a

generalised way.

2.) The subsubmodel GRID_TRAFO provides the interface routines to use these grid information for the transformation between the different grids. GRID_TRAFO utilises third party grid transformation codes: currently NREGRID[10] (Jöckel, 2006) and SCRIP (Jones, 1999).

The GRID User Manual in the Supplement provides detailed information about the usage of the GRID submodel.

---

[9]The names of MESSy submodels are written in capital letters throughout the article, even though they are not necessarily acronyms.

[10]Note: The infrastructure submodel previously used in EMAC is named NCREGRID, while the remapping algorithm itself is called NREGRID.

## 3.1 The SubModel Core Layer of GRID

Earth System Models usually define grids in spherical geometry. Two different grid types are distinguished in GRID v1.0: (1) rectangular grids, which are orthogonal in geo-coordinates and (2) curvi-linear grids. The implementation of non-rectangular, structured grids (e.g. the ICON grid) is ongoing.

Most of the internal data types of GRID follow the netCDF data model. The hierarchical data structures follow mainly those of NCREGRID (Jöckel, 2006). The definition of the Fortran structure, which contains all components required for the definition of a geo-hybrid grid, was extended and generalised for the usage in GRID. The geo-hybrid grid, as defined by Jöckel (2006), consists of a horizontal grid space, which comprises geographical latitude and longitude of the mesh vertices and / or centers. For different types of grids, different structure components for the definition of the horizontal grid are specified. The vertical

grid space is defined in analogy to the hybrid pressure level definition. Depending on the setting of the coefficients and of the reference and surface pressure, the vertical axis can be defined as one of (1) pressure hybrid pressure axes, (2) constant pressure axes, (3) constant height axes, or (4) sigma levels. More details can be found in the GRID User Manual in the supplement.

     The GRID SMCL routines also comprise subroutines for the handling of the grid structures, i.e., routines for initialising, copying, importing, exporting and printing a variable of the grid structure type. Beyond that, routines necessary for defining

a grid, storing it in a concatenated list, locating an already defined grid within this list, and for comparing grids, are part of the GRID SMCL. During a model simulation, the definition of an arbitrary number of geo-referencing grids and the transformations between those grids are possible.

### 3.1.1 GRID_TRAFO

The main intention of the GRID_TRAFO submodel is to provide routines for the transformation of gridded geo-located data.

GRID_TRAFO comprises NREGRID (Jöckel, 2006), the standard remapping tool in EMAC, and the SCRIP[11] software (Jones, 1999). While NREGRID is restricted to mapping between orthogonal 2-D or 3-D grids, SCRIP provides transformations to / from curvi-linear or unstructured grids. Here, we use grid "transformation" as generic term for both, conservative remapping (or "regridding") as well as for (not necessarily conservative) interpolation.

     The geo-hybrid grid structure provides all information required for the grid conversion. As each remapping software (NRE-

25 GRID and SCRIP) relies on its specific grid information structure, GRID_TRAFO additionally provides routines to extract these as required by the respective mapping software, i.e., it provides the "middleware" or acts as "wrapper" for the established mapping software. The remapping algorithms automatically apply the correct conversion routines, depending on the associated structure components. While the core mapping algorithms differ, GRID_TRAFO provides unified interfaces for the conversion between different grids. Additional interpolation schemes can be easily added, if required in the future.

The details are explained in the GRID User Manual, which is part of the Supplement.

---

[11] *S*pherical *C*oordinate *R*emapping and *I*nterpolation *P*ackage

### 3.1.2 NREGRID

The remapping algorithm NREGRID is a recursive algorithm, which is applicable to arbitrary orthogonal (including curvilinear) grids of any dimension. It is used for the rediscretisation of "gridded" geo-scientific data between n-dimensional (usually n = 2 or 3) orthogonal grids. The conservative rediscretisation of extensive or intensive variables is based on the calculation of the overlap (area or volume) matrix between source and destination grid boxes. For orthogonal grids these overlap matrices can nicely be calculated recursively, since the overlap area / volume is zero as soon as at least the overlap interval along one axis (dimension) is zero. Since the recursive nature of this algorithm limits its application to orthogonal grids, it cannot be applied for rediscretisations between the (in geographical coordinates) orthogonal Gaussian grid of ECHAM5 and the rotated (in geographical coordinates non-orthogonal) COSMO grid.

Details about the NREGRID algorithm have been published by Jöckel (2006).

### 3.1.3 SCRIP

As NREGRID is limited to the remapping between equally oriented orthogonal grids, the implementation of an algorithm able to transform between different curvi-linear or even unstructured grids became necessary. For this, the SCRIP software[12] (Jones, 1999) version 1.4 provided by the Los Alamos National Laboratory has been utilised. SCRIP (a *S*pherical *C*oordinate *R*emapping and *I*nterpolation *P*ackage) *"is a software package used to generate interpolation weights for remapping fields from one grid to another in spherical geometry. The package currently supports four types of remappings. The first is a conservative remapping scheme that is ideally suited to a coupled model context where the area-integrated field (e.g. water or heat flux) must be conserved. The second type of mapping is a basic bilinear interpolation which has been slightly generalized to perform a local bilinear interpolation. A third method is a bicubic interpolation similar to the bilinear method. The last type of remapping is a distance-weighted average of nearest-neighbor points. The bilinear and bicubic schemes can only be used with logically-rectangular grids; the other two methods can be used for any grid in spherical coordinates."* (Quoted from: SCRIP Users Guide, Introduction, Jones, 1998).

### 3.1.4 Application of GRID in MMD2WAY_CHILD

The COSMO model uses a rotated grid and the orientation between two COSMO model instances or the COSMO and the EMAC model is arbitrary. As NREGRID requires equally oriented orthogonal grids, it is not applicable in MMD2WAY. Sadly, SCRIP provides only algorithms for horizontal grid transformation. Thus two steps are required for the remapping of 3-D data fields.

Usually the biggest challenge in 2-way nesting of two atmospheric models, is the height correction required due to the differently resolved orographies of the child and the parent model. Thus, it seems to be a natural choice to first regrid horizontally and to perform the vertical regridding intertwined with the height adjustment as a second step. For child-to-parent coupling,

---

[12]http://oceans11.lanl.gov/trac/SCRIP (last access: 18 October 2017). The official link named in the SCRIP users guide (http://climate.acl.lanl.gov/software/SCRIP) is not available anymore.

first horizontal remapping via SCRIP is conducted. In a second step, the vertical remapping is performed using NREGRID for COSMO-EMAC coupling. For the COSMO-COSMO coupling, it was decided to use the INT2COSMO spline-interpolation.

Offhandedly, one might expect that INT2LM could be completely replaced by GRID, but this is not trivial. INT2LM provides much more functionalities than only remapping. It reads and processes the external data required as input for the COSMO model (especially for the initialisation of the model). Moreover, it performs some field adjustments concerning inconsistencies between the land-sea-mask of the COSMO model and the in-coming data. Therefore, it is not possible to completely eliminate INT2LM. One could, however, indeed exchange the horizontal and vertical interpolation routines. We started to test this, but in the first place the performance (w.r.t. the results) of the child model was downgraded. The main reason is that INT2LM does not only perform a vertical remapping, but preserves structure and characteristics of the boundary layer (i.e. up to 850 hPa), by moving it to the height of the target orography and remapping only the remaining part of the vertical column. This procedure, as it is implemented in INT2LM, is not reversible and thus introduces spurious effects for different orographies, which are always present because of the different horizontal resolutions of the nested models. Anyhow, for the off-line nested COSMO model this is the preferred way, as this makes physically more sense compared to a simple vertical interpolation. Unfortunately it causes inconsistencies, as it is not reversible. The supplement contains an example illustrating the deviations caused in the tracer profiles due to this height adjustment procedure.

## 3.2 The BaseModel Interface Layer of GRID

The backbone of each model is its grid, e.g., for an atmospheric model, the horizontal domain is given by a definition of the geographical longitudes and latitudes of the models grid midpoints and the grid corners. The vertical dimension is usually defined by a height or pressure based coordinate. As this grid (hereafter denoted as "basegrid") is the reference for most submodels and processes, the basegrid is defined in the basemodel interface layer (BMIL) for the usage in all MESSy submodels. In case of MMD2WAY, MMD2WAY_CHILD utilises the basegrid as source grid for the mapping to the "out-grid" as target grid.

## 4 Example Applications using the 2-way coupled MECO(n) system

Keeping the remaining issues in mind (Sect. 3.1.4), the current implementation, nevertheless, allows already for some useful applications. For instance, data can be transferred on-line from the finer to the coarser grid to be compared on the coarse grid. A simple example is shown in Sect. 4.1. Additionally, diagnostic tools can be used to interpret global and regional model results consistently, e.g. for radiative forcing, which is consistenty determined only, if calculated with the same radiation code. Section 4.2 illustrates this utilising the radiative forcing calculations of EMAC.

As discussed in Sect. 2.2, the 2-way coupling of prognostic variables is technically implemented in the MMD2WAY submodel. Thus Sect. 4.3.1 gives an example for an EMAC - COSMO/MESSy coupling, where dust tracers are coupled 2-way. Finally, Sect. 4.3.2 shows the full dynamically 2-way coupling of two COSMO/MESSy model instances located over the Atlantic ocean (i.e., over flat terrain) using the same height coordinates.

## 4.1 Simple examples of added value through aggregated subgrid-scale information

Depending on their resolution, only certain processes can be resolved by atmospheric models, whereas others have to be parameterised. Naturally, smaller scale models can resolve more processes explicitly. It is still under debate, whether or not the aggregation of the subgrid-scale information provided by the smaller scale model to the larger scale model constitutes an added value for the larger scale model. This issue might be answered with the help of 2-way coupled applications. Most probably, the answer will differ for different processes. For some dynamical processes, e.g. the generation of Rossby waves or Hurricanes (see Sect. 4.3.2), the upscaling might result in an added value, as these phenomena originate from smaller scale perturbations.

For chemistry models, especially the treatment of emissions is of interest. On the one hand, emissions, which depend on soil properties and/or on prognostic variables in the model (the so-called on-line emissions, because they are calculated during the simulation), can substantially differ between models with different resolution. One example are dust emissions, which depend on the 10m wind speed, soil properties and soil moisture (see Sect. 4.3.1). On the other hand, it is normally assumed, that even point and line emissions are instantly mixed within the grid box into which they are emitted. This leads to a higher dilution in larger scale models. Especially in highly polluted regions, or more generally near emission sources, this might influence the simulated chemical regime, as atmospheric chemistry is highly non-linear.

Figure 5 illustrates, as a simple example, the resolution effect on on-line calculated nitrogen oxide (NO) soil emissions (Kerkweg et al., 2006b). These emissions strongly depend on the soil properties and thus differ substantially between the models.

Panel A depicts the NO emission flux as calculated on a global EMAC model grid of T42 ($\approx 2.8°$) resolution. Panel B shows how these emission fluxes look on a COSMO/MESSy grid with $0.36°$ horizontal resolution. If COSMO/MESSy were 2-way coupled into EMAC and EMAC were using the NO emissions coupled from COSMO/MESSy instead of calculating them itself, the emissions aggregated from the COSMO/MESSy to the EMAC grid would be as in Panel C. Panel D depicts the difference in percent between the emissions directly calculated by EMAC (Panel A) and coupled back from COSMO/MESSy (Panel C). Naturally, the emission fluxes on the COSMO/MESSy grid show much finer structures as a result of the finer grid and therefore finer distributed soil properties. However, the largest differences between the up-scaled (Panel C) and in EMAC calculated (Panel A) emission fluxes occur at the coast lines (Panel D), which is mostly due to the much finer resolved land-sea mask in the smaller-scale model. The NO emission flux integrated over the coupled domain is $3.29 kg(NO)/s$ and $2.63 kg(NO)/s$ for the parent and the child model, respectively. Thus, the differences in the soil properties of the two models account for a difference of $0.66 kg(NO)/s$. The integrated NO emission flux regridded from the child to the parent grid is $2.78 kg(NO)/s$, providing an emission flux lower by $0.51 kg/s$ compared to the directly calculated integrated emission flux. The difference of $0.15 kg(NO)/s$ between the flux in the regional domain and its integral over the global domain simply results from the not fully congruent areas, over which the integrals are taken in the rotated domain and the global domain, respectively.

As a second example, the dry deposition velocities for ozone are displayed in Fig. 6 (Kerkweg et al., 2006a). The features discussed for the previous example appear here as well. Additionally, the ozone dry deposition velocities calculated by COSMO/MESSy are much more evenly distributed in the Mediterranean region, while they are sligthly but systematically

smaller over Eastern Europe, which is most propably due to different soil properties and also due to the different turbulence schemes employed by the two basemodels.

## 4.2 Use of specific diagnostic tools: radiative forcing

To evaluate the radiative forcing for two MECO(n) instances consistently, the MESSy submodel RAD (Dietmüller et al., 2016) is used for the calculation of the radiative forcing of a COSMO/MESSy instance on-line coupled to the EMAC model. As COSMO/MESSy and EMAC use different radiation schemes, this is one way of a consistent comparison.

Here, results are shown from simulations using a setup as published by Mertens et al. (2016). A COSMO/MESSy instance over Europe ($0.44°$ resolution) was coupled to the global EMAC domain. The ozone field calculated by COSMO/MESSy was sent back to EMAC using MMD. However, the ozone field coupled back from COSMO/MESSy is zero or undefined outside of the coupled region, i.e., the horizontal and vertical relaxation areas as well as those parts of the globe, which are not covered by the COSMO/MESSy instance. Therefore, the uncovered points are filled with the ozone field calculated by EMAC, as for the calculation of the radiative flux in EMAC, global, non-zero fields must be fed into the diagnostic routine. With this ozone field a second, diagnostic radiation call is performed using RAD. Two simulations are investigated:

- **REF**: EMAC and COSMO/MESSy are using the same emission data set (MACCIty, Granier et al.,2011).

- **SENS**: EMAC uses the MACCIty inventory, while COSMO/MESSy applies a DLR specific inventory.

Here, the difference ('COSMO/MESSy minus EMAC') of the radiative fluxes, area-averaged over Central Europe ($35°$:$60°$ N; $-10°$:$30°$ E) for July 2008, are compared. Figure 7a shows the vertical profiles of the differences of the clear-sky radiative fluxes applying the same emissions (REF). The larger ozone values as simulated by COSMO/MESSy compared to EMAC near the tropopause lead to a positive radiative flux difference in the longwave as well as in the shortwave bands around 200 hPa. If the emissions in COSMO/MESSy are changed (SENS, Fig. 7b), lower ozone values are simulated by COSMO/MESSy compared to EMAC up to around 800 hPa. These lower values lead to a negative difference of the longwave radiative fluxes compared to EMAC.

## 4.3 2-way coupling of prognostic variables

Next, we show two examples for the coupling of 3-D prognostic variables. First, the 2-way coupling of dust tracers between EMAC and one COSMO/MESSy instance is shown. Secondly, all dynamical variables of two COSMO/MESSy instances are coupled 2-way to demonstrate the potential of the 2-way coupling. To avoid the adjustment of the orographies, the smaller COSMO/MESSy instance is predominantely located over the ocean.

### 4.3.1 Dust

Dust emissions are very sensitive to the model resolution, as they depend on the soil type and the wind velocity. Typically, dust emission schemes are developed for a specific model resolution. They include scaling factors to adapt them as well as possible to other resolutions.

In our example, we use a MECO(1) setup, coupling the dust tracers of a COSMO/MESSy instance with 0.36° horizontal resolution back to the EMAC model in T63 spectral resolution.

Figure 8 shows the dust emission fluxes integrated over a domain ranging from 60°W to 60°E and from 45°S to 45°N. Due to different soil type distributions and higher wind maxima in the COSMO/MESSy instance, the latter produces much higher dust emission fluxes, as the simulation was performed without any resolution dependent tuning of the emission scheme. These higher emissions are reflected also in the horizontal dust column mass (mg/m$^2$) distribution. Figure 9 displays the dust column mass for March 06, 2004. Panel A shows the dust column mass (in mg/m$^2$) in the COSMO/MESSy instance, panel B the EMAC dust column mass in the 1-way coupled simulation, and panel C the result of the 2-way coupled simulation. Obviously, the COSMO/MESSy instance exhibits much finer structures as both EMAC instances. However, the maximum present in the COSMO/MESSy instance is much better represented in the EMAC 2-way simulation, as intended.

This simulation contains still a small error with respect to the vertical distribution of the dust, as the height adjustment for the orography is not yet consistent for the two coupling directions (see Sect. 3.1.4). Nevertheless, this example illustrates the potential of the 2-way coupling to improve the coarse representation of quantities, which are determined by smaller scale features.

### 4.3.2 Hurricanes

Tropical cyclones (TCs), developing at the West coast of Africa over the tropical East Atlantic, are known to be precursors for hurricanes causing damages in the US (e.g., Ike, 2008; Dean, 2009) or over Europe (e.g., Helene, 2006; Katia, 2011). Those TCs often originate as disturbances of the African Jet Stream, so-called African Easterly Waves (AEWs) over the African continent. In case of suitable conditions over the Atlantic, these AEWs have the potential to develop into TCs and finally to hurricanes. Therefore, forecasting the development, track and intensity of hurricanes requires both, a high model resolution to capture all the multi-scale interactions, prerequisite for the development of the initial TC, and a huge model domain capturing the African continent as well as the Atlantic ocean (e.g., Rappaport et al., 2009; Schwendike and Jones, 2010).

As an example, the development of a hurricane named ISAAC is analysed here. It originated in September 2000 as a TC from an AEW at the West coast of Africa. The National Hurricane Center classified it as a hurricane for the first time on 23 September 12 UTC, before it reached maximum intensity as a category 4 hurricane on 28 September 18 UTC (https://coast.noaa.gov/hurricanes/). Afterwards, its track turned to the north-east and after extratropical transition the system reached Great Britain readily identifable by strong wind gusts two days later.

To demonstrate the potential of the dynamical 2-way coupling between two COSMO/MESSy instances, a MECO(2) set-up is applied to simulate the development of ISAAC. This means, two COSMO/MESSy instances, varying in horizontal res-

olution, time step length and model domain, are coupled to the global EMAC model. The finer resolved ($0.11° \approx 12$ km) COSMO/MESSy instance is driven by the coarser resolved ($0.22° \approx 25$ km) COSMO/MESSy instance. Initial and boundary data for the coarser COSMO/MESSy instance are transformed from EMAC (T106 $\approx 120$ km). Since this study focuses on the specific development of ISAAC, a weak nudging of four prognostic variables (temperature, divergence, vorticity and the logarithm of surface pressure) towards ECMWF analysis data is applied for EMAC (as described by Jöckel, 2006) during the first two weeks after start of the simulation (15 September 0 UTC). Once the hurricane leaves the model domain of the finer resolved COSMO/MESSy instance (29 September 0 UTC, Fig. 11) the nudging is switched off and the EMAC instance is completely unconstrained afterwards. In case the COSMO/MESSy instances are coupled 2-way, the dynamical information from the finer resolved instance, comprising the temperature (T), the wind velocities (U, V, W), the pressure deviation from the reference atmosphere (PP) and moisture (QV, QC, QI) are fed back to the coarser instance. The domains covered by the model instances are shown in Fig. 11 (grey and blue areas).

In Fig. 10 and 11 the results obtained with the 2-way coupled COSMO/MESSy instances (right panels) are compared to those of the 1-way coupled system (left panels). Deviations are validated using the HURDAT data set (Landsea and Franklin, 2013), which is part of the International Best Track Archive for Climate Stewardship (Knapp et al., 2010, IBTrACS (v03r04), available under doi:10.7289/V5NK3BZP).

Although EMAC is nudged during the genesis phase, the development of ISAAC is not captured by EMAC, as there is no pressure decrease visible in the time series of minimum sea level pressure ($SLP_{min}$, Fig. 10, black contour). In contrast, ISAAC initially originates in both COSMO/MESSy instances (Fig. 10, blue and red contours), independent of the coupling strategy and horizontal resolution and approximately at the correct time (23 September) compared to the best track estimate (Fig. 10, green contour). However, there are strong differences comparing the 1-way and 2-way coupled instances during the ongoing development of $SLP_{min}$: while the final intensification of ISAAC simulated with the 1-way coupled instances does not start before 26 September, the 2-way coupled instances are able to capture the initial decrease. Even though the intensity of ISAAC in the 1-way coupled simulations coincides better with the reference on 27 September, the position and further track of ISAAC differs distinctly from the best track position (Fig. 11) from this time on. In contrast, the dynamical 2-way coupling between the COSMO/MESSy instances leads to a correct representation of the track and intensity of ISAAC in the coarser resolved model instance, even after the system has left the model domain of the finer resolved instance.

By simulating the development of ISAAC with the MECO(2) set-up, the potential of the dynamical 2-way coupling between two COMSO/MESSy instances is demonstrated: to capture the multi-scale interactions, prerequisite for the development of the initial TC of ISAAC, in this case a horizontal model resolution of $0.11°$ is required. The model domain of this fine resolved instance, however, can be kept small, if the dynamical information are fed back to the coarser resolved model instance in the 2-way coupled mode.

Overall, the results of the examples shown here, indicate that the 2-way coupling has the potential to improve the representation of hurricanes in the coarser COSMO/MESSy model instance.

## 5 Model Performance

Due to technical reasons, the frequency of data exchange between the child and the parent model must be the same as for the parent-to-child data transfer. During the latter, the two time slices of the boundary fields, between which COSMO usually performs a linear time interpolation, are now filled with the data of the actual time step. This was required to enable a non-sequential 2-way coupling. However, it limits the choice of the coupling frequency, which should be chosen as small as possible, i.e., as the smallest common multiple of the parent and the child model time step. For this reason, a sensitivity analysis of different coupling frequencies is not provided here.

Usually people are aware of other couplings (e.g. ocean-atmosphere coupling) in which, for instance for mass conservation, fluxes need to be accumulated / averaged over the coupling interval. In contrast to this, our 2-way coupling of two atmosphere models utilises a relaxation technique at the lateral boundaries for the parent-to-child exchange. For the child-to-parent coupling a relaxation for the entire coupling domain modifies the coarser model results according to the finer resolved fields. Thus, since we do not couple fluxes for which mass conservation would be required, but correct the results directly, accumulation or averaging over time is not feasible.

The run-time performance depends first and foremost on the specific model setups (e.g. on the complexity of the chosen chemistry representation etc.). But in the end, the overall performance is mostly determined by the "degree of balance" of the distribution of parallel tasks among the different model instances. We discussed this in detail in Part 2 of our series (Kerkweg and Jöckel, 2012b) for the 1-way nesting case. The same principles hold for the 2-way exchange, except for the complication that communication waiting times depend now on bi-directional data exchange. Thus, it is up to the user to find (experimentally) the optimum task distribution to minimise communication waiting times. The supplement contains an example, showing the additional costs of 2-way coupling, for one specific MECO(2) setup without chemistry.

Fig. 12 sketches exemplarily the costs of the coupling. A MECO(2) setup similar to the hurricane case was integrated for 1 day and the residence time in the respective routines transforming the data has been measured. Because the child model does all the data transformations between the two grids, it consumes much more computing time than the parent model. The difference between the 1-way coupled (black) and the solely dynamically 2-way coupled (red) simulation is small, as only six additional fields need to be interpolated. Already for the 1-way coupled simulation, adding 139 chemical tracers (black dashed line) triples the processing time in the child model, while it requires the sixfold time, if they are 2-way coupled (red bashed line).

In contrast to this, the number of coupling fields provokes no systematic increase of computing time in the parent model. Although the coupling time increases significantly in the child model, the time consumption for the coupling is still negligible in comparison to the computing time required for the calculation of the chemistry of these 139 chemical tracers.

## 6 Conclusions

In this article we present the next generation of the Multi-Model-Driver (MMD v2.0). While MMD (v1.0) provided all tools required for the 1-way coupling of dynamical and chemical models (e.g., EMAC and an arbitrary number of COSMO/MESSy

instances) following a client-server approach, version v2.0 was further developed to allow for data exchange from the client to the server model.

To reach this goal, the MMD library was expanded by the respective subroutines for the data exchange via MPI. The new submodel MMD2WAY includes the features of the previous MMD (v1.0) submodels MMDCLNT and MMDSERV plus all

functionalities for the data transfer from the child to the parent model.

The new MESSy infrastructure submodel GRID is used for the transformation from the child model grid to the subpart of the parent model grid overlapped by the child model. For the horizontal data remapping the SCRIP software implemented in the GRID submodel is used. For the vertical regridding for COSMO-EMAC coupling NREGRID, which is also part of the GRID submodel, is utilised. In contrast to this, the vertical remapping for the COSMO-COSMO coupling is performed using

the spline-interpolation as provided by INT2COSMO.

Currently, an inconsistency between the vertical regridding routines in INT2COSMO and GRID, and in the adaption of the resolution dependent orography limits a fully consistent dynamical coupling. This is especially the case for the EMAC-COSMO/MESSy coupling, as the vertical coordinate of EMAC is pressure based, while the COSMO model uses a height based vertical coordinate. Nevertheless, over flat terrain and between COSMO/MESSy instances coupling of 3-D prognostic

variables is possible.

The capabilities of the 2-way coupling have been demonstrated on the basis of four examples: (a) a comparison of fields upscaled from the regional model to the global model grid with the global model fields, (b) the comparison of radiative forcing calculated consistently with the same radiation scheme, (c) the 2-way coupling of dust tracers, which emission fluxes are highly grid resolution dependent, and (d) the dynamical coupling of two COSMO/MESSy instances influencing the development of a

hurricane within the coarse COSMO/MESSy model domain.

The Supplement contains the manuals for the MESSy infrastructure submodel GRID, for the MMD Library and the MMD User Manual.

To develop a fully dynamical 2-way coupling for the MECO(n) system, the INT2COSMO routines need to be replaced by more generic routines to enable a fully consistent coupling in both directions.

*Code availability.* The submodel GRID and the 2-way coupling code are part of the official MESSy distribution (code release v2.53 and younger). The code as described is part of the Modular Earth Submodel System (MESSy), which is continuously further developed and applied by a consortium of institutions. The usage of MESSy and access to the source code is licenced to all affliates of institutions which are members of the MESSy Consortium. Institutions can be a member of the MESSy Consortium by signing the MESSy Memorandum of Understanding. More information can be found on the MESSy Consortium Website (www.messy-interface.org).

*Author contributions.* A. Kerkweg was the leader of the FLAGSHIP project. She developed the largest part of MMD (library and MMD2WAY submodel) and the generic submodel GRID. Ch. Hofmann and G. Pante have been part of the FLAGSHIP project and considerably contributed to the MMD2WAY development. Ch. Hofmann performed the hurricane study. G. Pante provided the results of the dust tracer

example. M. Mertens was the first "FLAGSHIP external user" of the 2-way coupled system and contributed the radiative forcing example. P. Jöckel was one of the master minds of the FLAGSHIP team. Additionally, the GRID software is based on the NCREGRID software developed by him. Thus he considerably contributed to the implementation of the GRID submodel.

*Acknowledgements.* The work was financed by the German Ministry of Education and Research (BMBF) in the framework of the MiKlip (Mittelfristige Klimaprognose / Decadal Prediction) subproject FLAGSHIP (Feedback of a Limited-Area model to the Global-Scale implemented for HIndcasts and Projections). Mariano Mertens acknowledges funding by the DLR project "Verkehr in Europa". We thank Bastian Kern and Andrea Pozzer for fruitful discussions concerning the application of SCRIP for interpolation between the EMAC and the MPIOM grid. The authors wish to acknowledge use of the Ferret program for the graphics in this paper. Ferret is a product of NOAAs Pacific Marine Environmental Laboratory (Information is available at http://www.ferret.noaa.gov). Additionally we thank the members of the MiKlip subproject MesoTel (Andreas Will, Ingo Kirchner, Markus Thürkow, Stefan Weiher and Mareike Schuster) for fruitful discussion of the dynamical 2-way coupling between the EMAC and the COSMO/MESSy model.

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

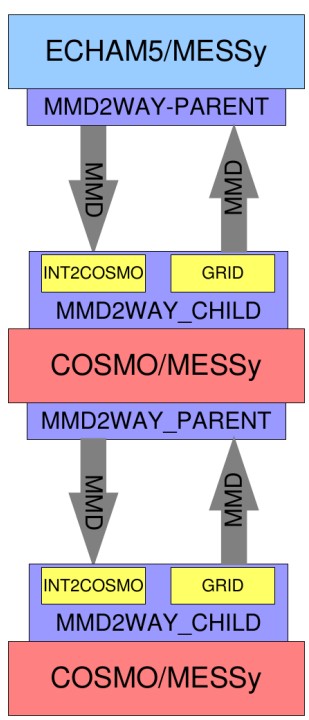

**Figure 1.** Illustration of the connection between the different MMD parts. The example is for a MECO(2) setup, i.e., EMAC with a cascade of 2 nested COSMO/MESSy instances.

Zaengl, G., Reinert, D., Ripodas, P., and Baldauf, M.: The ICON (ICOsahedral Non-hydrostatic) modelling framework of DWD and MPI-M: Description of the non-hydrostatic dynamical core, QUARTERLY JOURNAL OF THE ROYAL METEOROLOGICAL SOCIETY, 141, 563–579, doi:10.1002/qj.2378, 2015.

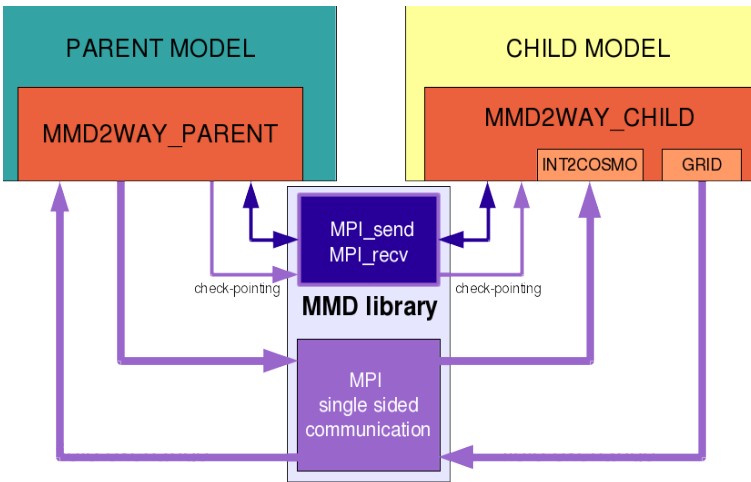

**Figure 2.** Illustration of the communication managed by MMD between a parent and a child model. Dark violet colours indicate data flow during the initial phase, while purple indicates the data flow during the integration phase.

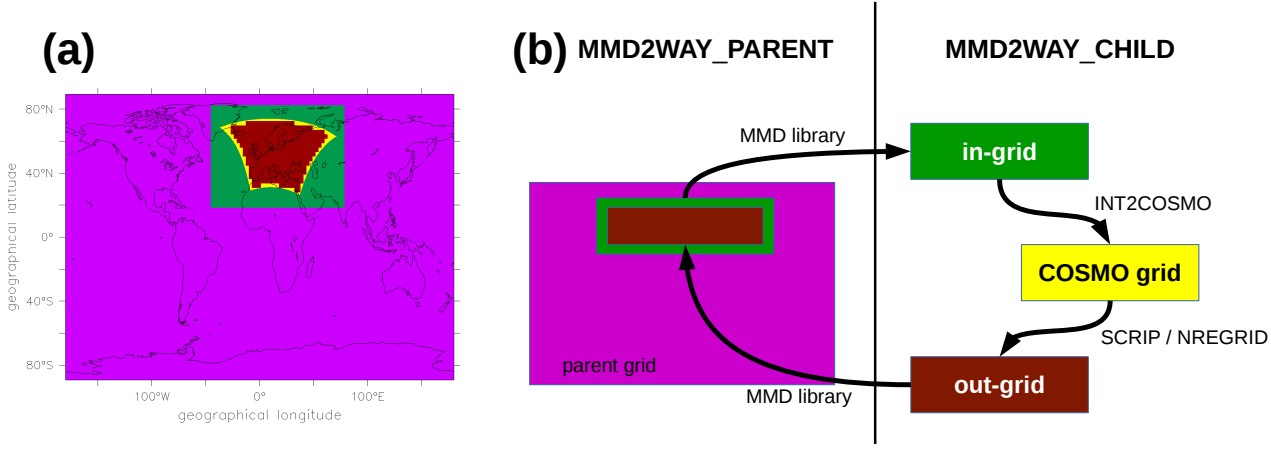

**Figure 3.** Relation of the different grids: EMAC grid in pink, the in-grid and out-grid defined by MMD2WAY_CHILD in green and brown, respectively, and the COSMO/MESSy model grid in yellow. Panel (a): position of the different grids relative to each other, for the example of a European domain; Panel (b) illustration of data flow between the different grids.

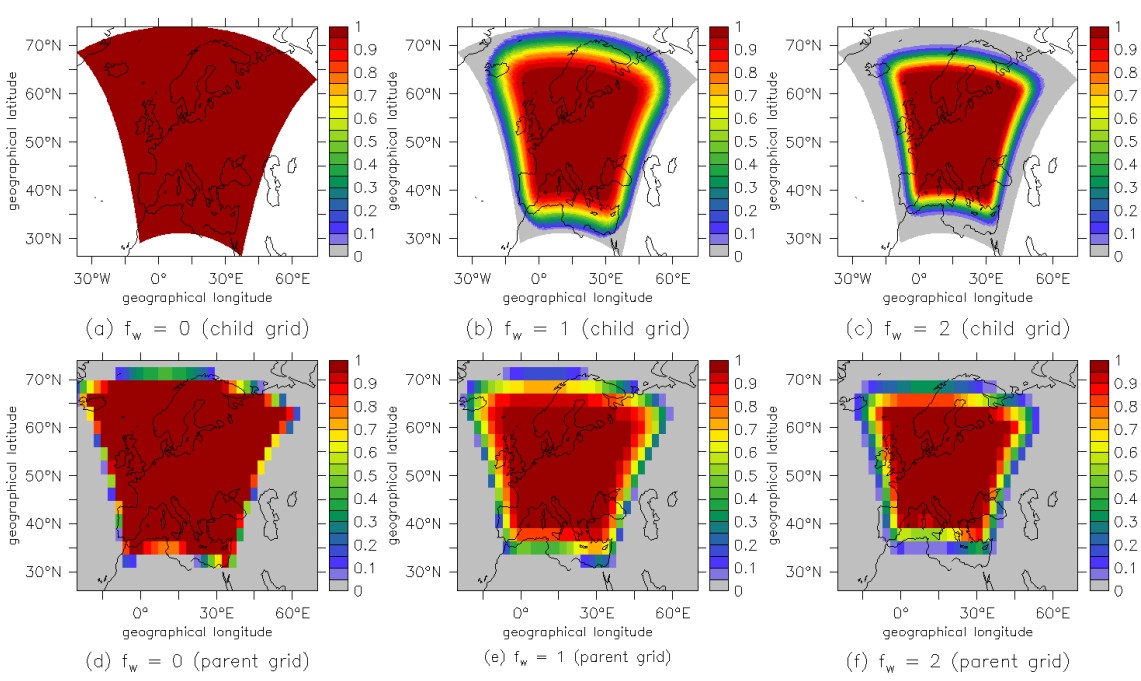

**Figure 4.** Weight functions ($f_w$, see Sect. 2.2, page 9) for the different weight types. Upper row: weight functions as calculated on the child model grid. Lower row: weight functions after transformation to the parent model grid.

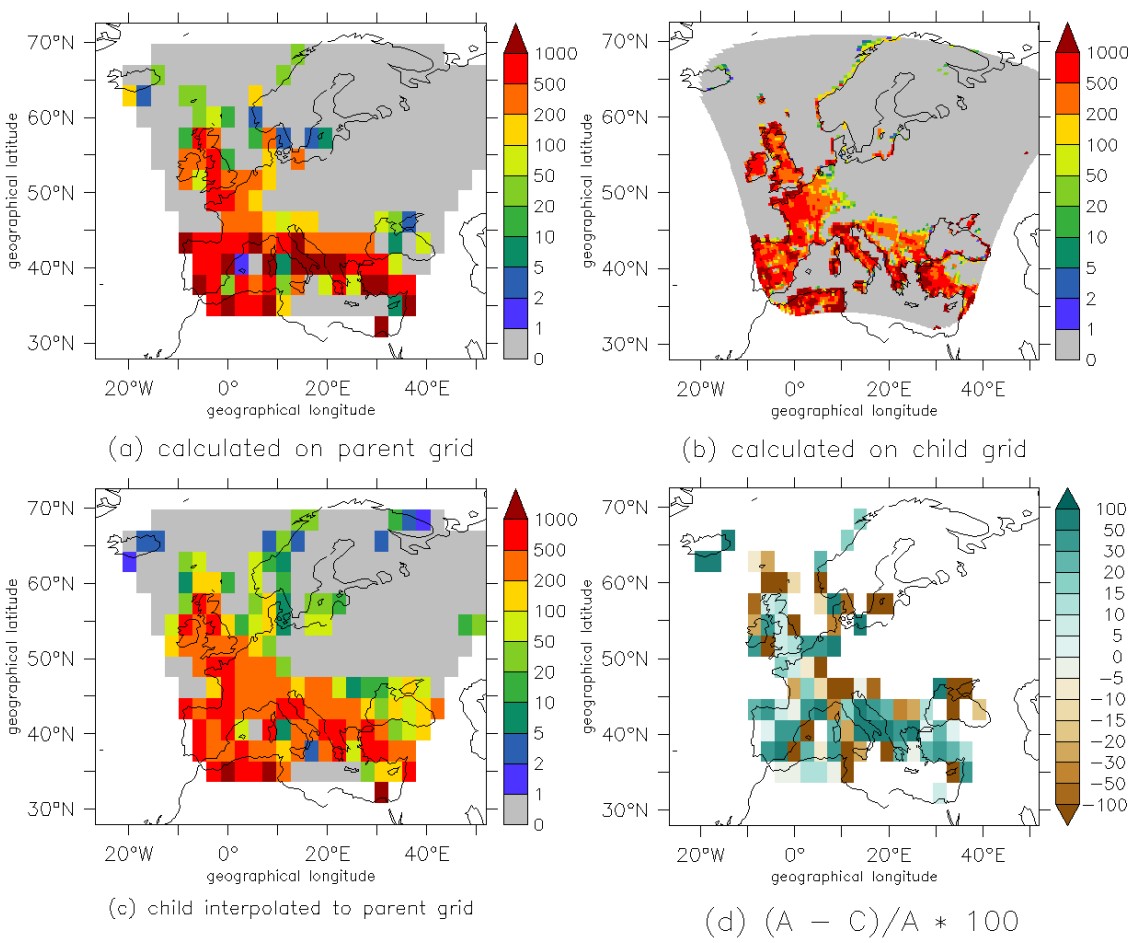

**Figure 5.** Biogenic NO emissions flux in $pg\ m^{-2}\ s^{-1}$ for one distinct date in January 2003. The data has been masked to show only the coupled region.

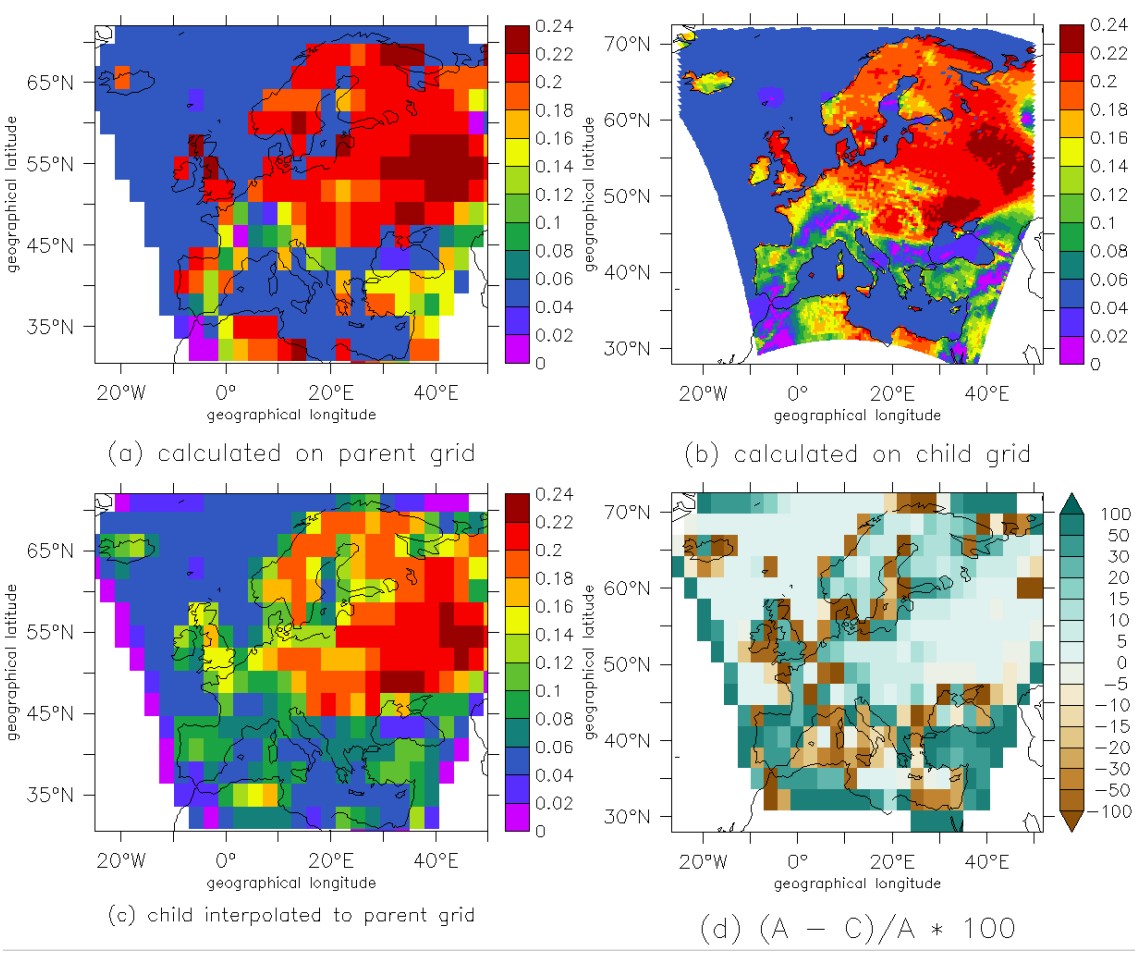

**Figure 6.** Ozone dry deposition velocities in $cm\ s^{-1}$ for one distinct date in January 2003. The data has been masked to show only the coupled region.

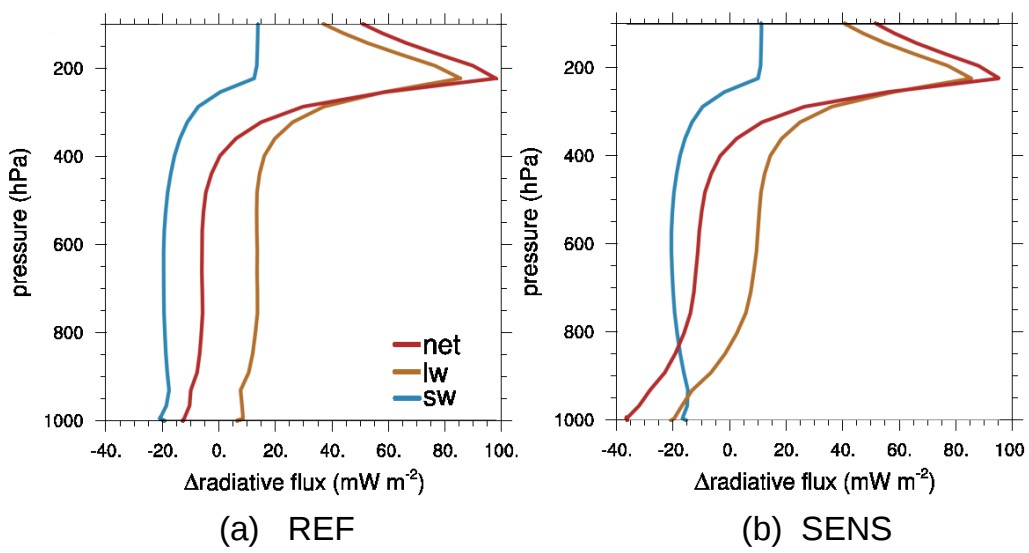

**Figure 7.** Differences ('COSMO/MESSy minus EMAC') of the radiative fluxes (shortwave (sw), longwave (lw) and net) averaged over $35°$N - $60°$ N and $-10°$E - $30°$ E for July 2008. (a) REF simulation; (b) SENS simulation.

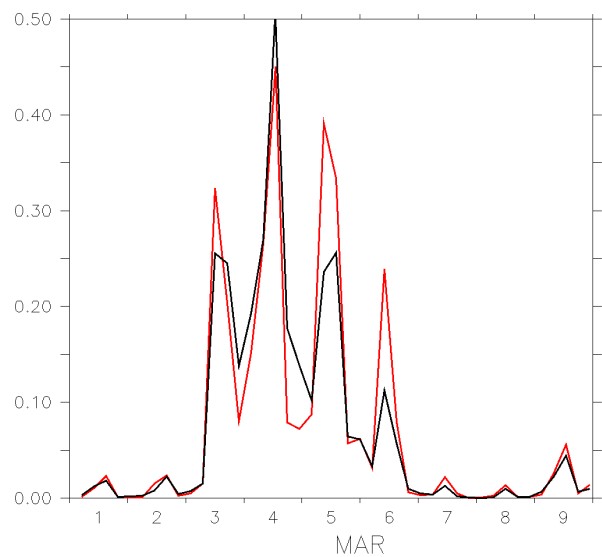

**Figure 8.** Time series of dust emission (Gg) in the EMAC (black) and in the COSMO/MESSy (red) instance.

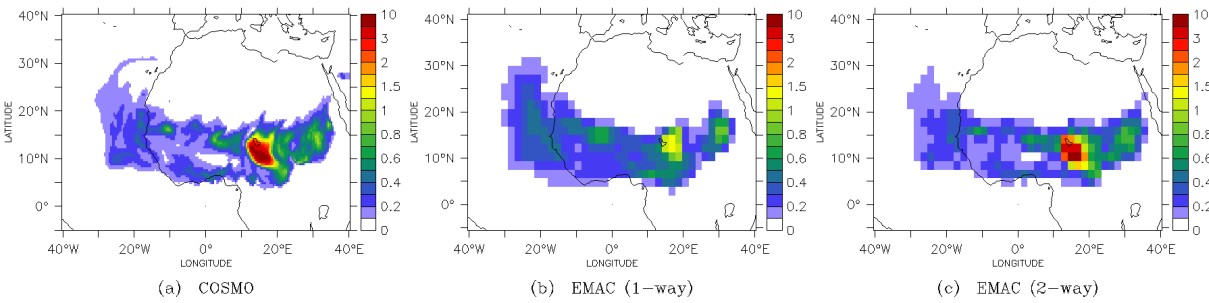

**Figure 9.** Dust column mass (mg $m^2$). Shown is an instantaneous value for March 06, 2004, 00 UTC. Left panel: for COSMO/MESSy; middle panel: for EMAC; right panel: for 2-way coupled EMAC, thus influenced by the COSMO/MESSy instance.

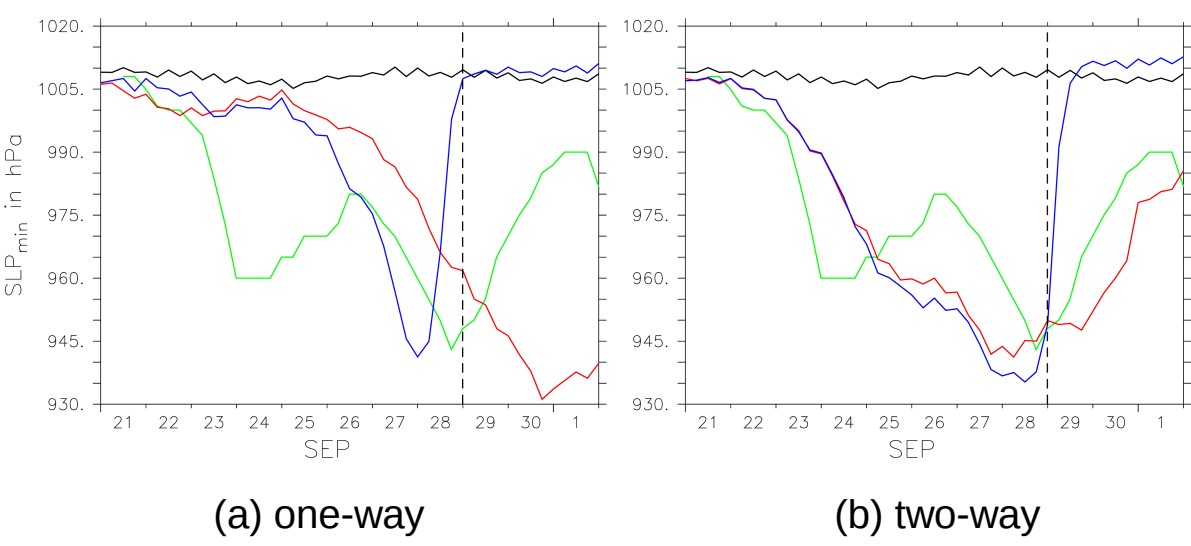

**Figure 10.** Time series of SLP$_{min}$ (in the area of 10°N - 50°N and 65°W - 25°W) in the one-way (left) and 2-way (right) coupled simulation for EMAC (black), COSMO/MESSy$_{0.22}$ (red), COSMO/MESSy$_{0.11}$ (blue) based on 6-hourly data. The best-track intensity from HURDAT is shown as reference (green). EMAC is nuged until 29 September (dashed line, s. text for details).

.

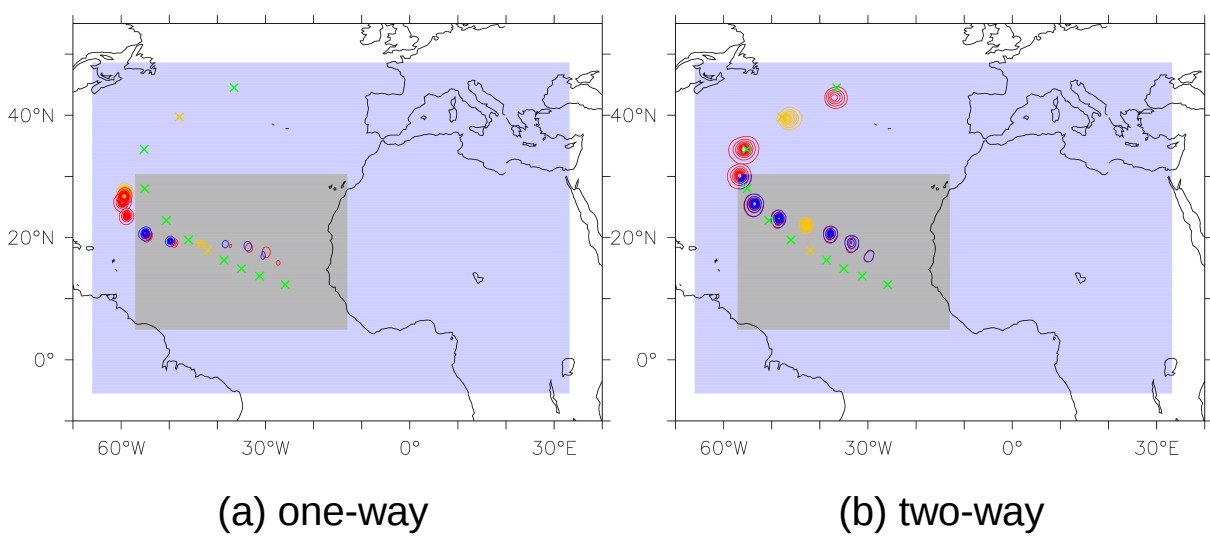

| (a) one-way | (b) two-way |

**Figure 11.** Track position for hurricane ISAAC in the one-way (left) and 2-way (right) coupled simulation. The daily SLP (less than 1005 hPa, 5 hPa-intervalls, in the area of 0°N - 50°N and 65°W - 25°W, starting on 23 September 0 UTC) is shown as contours for the COSMO/MESSy instances (red: $COSMO/MESSy_{0.22}$, blue: $COSMO/MESSy_{0.11}$). The best-track position from HURDAT is marked as reference (green crosses). To allow for a temporal comparison, the positions on 26 September and 1 October are yellow coloured for all tracks. The different model domains of the COSMO/MESSy instances are shaded (blue: $COSMO/MESSy_{0.22}$, grey: $COSMO/MESSy_{0.11}$).

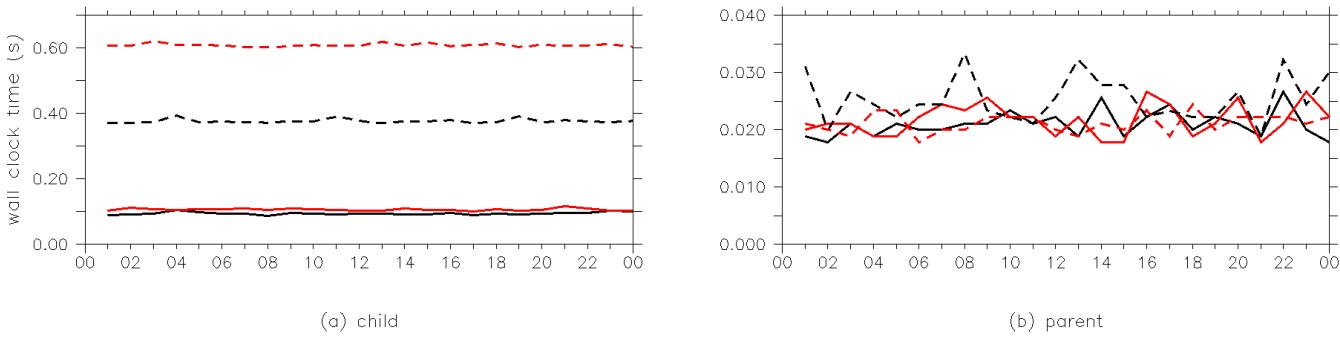

**Figure 12.** Hourly averaged wall clock time spent for the processing of the coupling data in the submodels (a) MMD2WAY_CHILD and (b) MMD2WAY_PARENT for a MECO(2) setup, shown for different couplings between the two COSMO/MESSy instances: one-way coupling (black), 2-way dynamical coupling (red). The dashed lines are for the additional coupling of 139 tracers.