# Peer review of "The on-line coupled atmospheric chemistry model system MECO(n) – Part 5: Expanding the Multi-Model-Driver (MMD v2.0) for 2-way data exchange including data interpolation via GRID (v1.0)"

_Geoscientific Model Development, 2017_

## Referee Comment (RC1) · Anonymous Referee #1 · 19 May 2017

Review of: "The on-line coupled atmospheric chemistry model system MECO(n) – Part 5: Expanding the Multi-Model-Driver (MMD v2.0) for 2-way data exchange including data interpolation via GRID (v1.0)"

By Astrid Kerkweg, Christiane Hofmann, Patrick Jöckel, Mariano Mertens, and Gregor Pante

Submitted to: GMD Manuscript gmd-2017-87

[Figure]

Recommendation: Minor revisions

Overview:

This manuscript documents new features of the MECO system (MESSyfied ECHAM and COSMO models nested). A new version (MMD v2.0) of the Multi-Model-Driver has been implemented and the capabilities of the 2-way coupling is illustrated.

General Comments: a) The manuscript is well written and the achieved model improvements are clearly described. b) A description of the time management during the 2-way coupling is missed. I would see a more detailed explanation in terms of coupling frequency, time slices considered to average (accumulate) fields before interpolation, etc. c) An evaluation of the MMD v2.0 model performances (the increased computational cost quantification, etc) compared to the v1.0 could improve substantially the present work.

Specific Comments: - Page 3 line 5: I suggest to uniform the syntax and to use coupling OR nesting throughout the paper. - Page 5 line 25: what do you mean with "longer simulation"? I assume this system as also available for climate simulations, thus a "restarting feature" is a mandatory requirement. Is it the system designed considering this feature? - Page 6 line 20: figure 3 labels (panel b) are not readable. Also please uniform the subpanel labelling [ a), b) .. ] in all of the manuscript figures. - Page 7 line 5: is there any plan to add other remapping approaches? - Page 7 line 10: you use "0. and 1." Instead of the "1. and 2." Approach used in the previous page. Why? - Page 8 line 15: figure 4 labels are not readable. - Page 9 line 1: "…as one central part" should be "..as the central part" - Page 9 line 4: What does "ideally" means? - Page 11 line 15: The remapping steps mentioned (first horiz. then vert.) are the typical ones. Not sure this is always the best way, depending on spatial resolution and fields considered. Is it possible to give the user the possibility to choose the interpolation order? - Page 12 line 10: the last sentence of this chapter is a conclusion before results description. I suggest to move it after the discussion of the TC example. -

[Figure]

Page 12 line 15: "For 2-way applications. . ..." please rephrase this sentence. - Page 12 line 25: "NO" must be typed explicitly. - Page 12 line 30: if I understand well, the only interpolation available is a conservative one. I suggest to add NO spatial integral values as obtained after and before the interpolation, to complement the information obtained by figure 5 and 6. - Page 13 line 30: I think it could help to see in the present work also the model deficiency induced by topography. - Page 14 line 10: what do you mean with "..performed without any scaling of the emissions" ? - Page 15: are we looking (figures 10 and 11) at daily or 6hourly (or model time step snapshot) values? Is it possible to see the same as figure 10, but based on 10 meter wind speed? - Page 15: I think it is really important to highlight the role of the coupling frequency when coupling components/models to improve the representation of certain features such as TCs (see Scoccimarro et al. 2017 and Zarzycki et al. 2016). Thus please add some comment on the coupling frequency you used and some information on sensitivity tests (if any). - FIGURES: Figure 1 and 2 can be also smaller: I suggest to leave more space to enlarge figures as figure 4. Labels are not readable in figure 3a, 4, 10 and 11. Please uniform subpanels labelling (also add it to figure 10 and 11. I suggest to set white colour for near 0 values in figures 3a, 4 and 9.

References mentioned in the review document: -Scoccimarro E., P.G. Fogli. K. Reed, S. Gualdi, S.Masina, A. Navarra, 2017: Tropical cyclone interaction with the ocean: the role of high frequency (sub-daily) coupled processes. Journal of Climate , doi: 10.1175/JCLI-D-16-0292.1

-Zarzycki, C. M., Reed, K. A., Bacmeister, J. T., Craig, A. P., Bates, S. C., and Rosenbloom, N. A.2016: Impact of surface coupling grids on tropical cyclone extremes in high-resolution atmospheric simulations, Geosci. Model Dev., 9, 779-788, doi:10.5194/gmd-9-779-2016.

———————————

---

## Author Comment (AC1) · 11 Jul 2017

Note: Referee comments are indicated in bold, answers are in regular blue font.

**Overview:**
**This manuscript documents new features of the MECO system (MESSyfied ECHAM and COSMO models nested). A new version (MMD v2.0) of the Multi-Model-Driver has been implemented and the capabilities of the 2-way coupling is illustrated. General Comments: a) The manuscript is well written and the achieved model improvements are clearly described.**
Thanks!

**b) A description of the time management during the 2-way coupling is missed. I would see a more detailed explanation in terms of coupling frequency, time slices considered to average (accumulate) fields before interpolation, etc.**
Usually people are aware of other couplings (e.g. ocean-atmosphere coupling) in which, for instance for mass conservation, fluxes need to be accumulated / averaged over the coupling interval. In contrast to this, our two-way coupling of two atmosphere models utilises a relaxation technique at the lateral boundaries for the parent-to-child exchange, and within the entire coupling domain for the feedback from child to parent, thus modifying the model results according to the finer resolved fields. Thus, since we do not couple fluxes for which mass conservation would be required, but correct the results directly, no accumulation or averaging is reasonable.
The coupling frequency can be changed per namelist, but to minimise the errors, it is strongly recommended to couple every parent model timestep. We add this information to the revised article within a newly added section "Model performance".

**c) An evaluation of the MMD v2.0 model performances (the increased computational cost quantification, etc) compared to the v1.0 could improve substantially the present work.**
Fig. 1 gives an impression of the costs of the coupling. A MECO(2) setup, similar to the hurricane case, was run for one day. During this simulation the wall clock time spent for the data transformation was measured using an internal tool utilising system clock counts. Because the child model does all the data transformations between the two grids, it consumes much more computing time than the parent model. The difference between the one-way coupled (black) and the only dynamically two-way coupled (red) simulation is small, as only six additional fields need to be transformed. Adding 139 chemical tracers to the one-way coupled setup (black dashed line) triples the processing time in the child model, while it requires the sixfold time, if they are two-way coupled (red dashed line).
In contrast to this, the number of coupled fields provokes no systematic increase of computing time in the parent model.
We add this to the revised article in a new section "Model performance".

[Figure]

[Figure]

Figure 1: Hourly averaged wall clock time, spent for the processing of the coupling data in the submodels (a) MMD2WAY_CHILD and (b) MMD2WAY_PARENT for a MECO(2) setup, for different couplings between the two COSMO/MESSy instances: Black: One-way coupling, red: two-way dynamical coupling. The dashed lines are for the additional coupling of 139 tracers.

**Specific Comments:**
**- Page 3 line 5: I suggest to uniform the syntax and to use coupling OR nesting throughout the paper.**
"Nesting" and "coupling" do not mean the same thing. The term "coupling" is much broader. In the context of model coupling, it describes the exchange of data between models (or components of those) in general. The term "nesting" is more specific. It describes the data transfer between two models of the same compartment (e.g. atmosphere), of which one typically resolves a larger domain and is used to drive the model with the smaller domain.
In order to explain the nature of our coupling correctly, we think, we need both terms in the abstract / introduction. In the remainder of the article (i.e. in two examples 4.2 and 4.3.2) we replaced the term "nested" by "on-line coupled" in order to unify the usage of nesting and coupling.

**Page 5 line 25: what do you mean with "longer simulation"? I assume this system as also available for climate simulations, thus "restarting feature" is a mandatory requirement. Is it the system designed considering this feature?**
"Check-pointing", which is the technical term for "restarting feature", is indeed considered for the reasons given in the text. We remove "For longer simulations" and rewrite simply "Check-pointing" (the technical term for restarting feature) is required (not only for climate simulations) to be able ...".

**Page 6 line 20: figure 3 labels (panel b) are not readable. Also please uniform the subpanel labelling [ a), b) .. ] in all of the manuscript figures.**
Labels are changed / updated.

**Page 7 line 5: is there any plan to add other remapping approaches?**
Yes. Especially, as SCRIP provides other horizontal remapping approaches, it is relatively straightforward to implement them as well. But there is no special need (and funding) to do so at the moment. Additionally, it is discussed whether to implement YAC (Hanke et al., GMD, 2016) into GRID.

**Page 7 line 10: you use "0. and 1." Instead of the "1. and 2." Approach used in the previous page. Why?**
Because the previous page gives just a list of steps which are processed, while here "0. and 1." are indeed the numbers, which can be set in the namelist to choose the "method". To avoid the confusion, we change the numbers on page 6 to bullet points and use quotes for the numbers on page 7.

**Page 8 line 15: figure 4 labels are not readable.**
Changed.

**Page 9 line 1: "as one central part" should be "as the central part"**
This seems to be a misunderstanding due to incorrect use of language. Grid transformation is only one of a number of important parts in a model infrastructure. Others are e.g., memory management or time and event handling. Therefore we rephrase to "one important part".

**Page 9 line 4: What does "ideally" means?**
"Ideally" means that in the best case, the GRID submodel provides all the listed functionalities. So far, not all of them are implemented. This will be clarifed in the revised article.

**Page 11 line 15: The remapping steps mentioned (first horiz. then vert.) are the typical ones. Not sure this is always the best way, depending on spatial resolution and fields considered. Is it possible to give the user the possibility to choose the interpolation order?**
With some considerable additional programming effort it would be somehow possible. However, as usually the biggest problem is the height correction required due to the differently resolved orographies of the child and the parent model, it seems to be a natural choice to first regrid horizontally and to perform the vertical regridding intertwined with the height adjustment as a second step. At the time being we are not convinced that the effort of code restructuring would be justified by the scientific gain. We explicitly state this in the revised article.

**Page 12 line 10: the last sentence of this chapter is a conclusion before results description. I suggest to move it after the discussion of the TC example.**
Done.

**Page 12 line 15: "For 2-way applications. . ..." please rephrase this sentence.**
We changed "For 2-way coupled applications the questions, if the aggregation of the subgrid-scale information provided by the smaller scale model to the larger scale model constitutes an added value for the larger scale model is still under debate." to "The question, if the aggregation of the subgrid-scale information provided by the smaller scale model to the larger scale model constitutes an added value for the larger scale model is still under debate and might be answered with the help of 2-way coupled applications."

**Page 12 line 25: "NO" must be typed explicitly.**
Done.

**Page 12 line 30: if I understand well, the only interpolation available is a conservative one. I suggest to add NO spatial integral values as obtained after and before the interpolation, to complement the information obtained by figure 5 and 6.**

The NO emission flux integrated over the coupled domain is $3.29\,kg(NO)/s$ and $2.63\,kg(NO)/s$ for the parent and the child model, respectively. Thus, the differences in the soil properties of the two models account for a difference of $0.66\,kg(NO)/s$. The integrated NO emission flux regridded from the child to the parent grid is $2.78\,kg(NO)/s$, providing an emission flux lower by $0.51kg/s$ compared to the directly calculated integrated emission flux. The difference of $0.15\,kg(NO)/s$ between the flux on the regional domain and its integral on the global domain simply results from the not fully congruent areas (due to different grid box sizes and orientation) over which the integrals are taken in the rotated domain and the global domain, respectively.
We add this information to the article.

**Page 13 line 30: I think it could help to see in the present work also the model deficiency induced by topography.**

Differently resolved topography heights in the coupled models cause a displacement of the tracer with height. To visualise these differences, a MECO(2) simulation with a passive tracer was performed. The initial tracer distribution is horizontally homogeneous and vertically increasing. Fig. 2 displays at four different locations, the height profiles of the tracer in the parent domain (black, triangles), in the child domain (blue, circles) and the coupled field (red, upside down triangles). The annotation gives the surface height in the parent and the child domain, respectively. The blue and the black line are always on top of each other indicating the tracer is initialised with exactly the same height profile in both COSMO instances. With increasing surface height difference, the difference in the vertical profiles increases. The second row of Fig. 2 displays the differences of the black and the red line, i.e., of the original profile and the profile given by the coupling field.

We will add this explanation to the supplement of the paper and add a reference to the supplement to the paper.

**Page 14 line 10: what do you mean with "performed without any scaling of the emissions" ?**

Dust emission schemes heavily depend on soil properties, soil wetness and wind speed. All these factors vary with model resolution. Therefore, our dust emission scheme needs to be optimised by scaling the simulated flux for a given horizontal resolution, in order to yield the same integrated emitted dust mass. In this example we used the same scaling factors as for the global model in T42 also for the regional model.
We change the sentence to "without any resolution dependent optimisation of the emission scheme"

**Page 15: are we looking (figures 10 and 11) at daily or 6hourly (or model time step snapshot) values? Is it possible to see the same as figure 10, but based on 10 meter wind speed?**

These are 6 hourly values. We add this to the caption of the figure.
Fig. 3 shows the maximum 10m wind speed. As these figures do not provide any additional insights, we are hesitating to include them into the revised manuscript.

**Page 15: I think it is really important to highlight the role of the coupling frequency when coupling components/models to improve the representation of certain features such as TCs (see Scoccimarro et al. 2017 and Zarzycki et al. 2016). Thus please add some comment on the coupling frequency you used and some information on sensitivity tests (if any).**

Due to technical reasons, the frequency of data exchange between the child and the parent model must be the same as for the parent-to-child data transfer. For the latter, the two slices of the boundary fields, for which COSMO performs a linear time interpolation, are filled with the data of the actual time step of the parent model. This was required to enable the two-way coupling in which parent and child instances are running concurrently (and not sequentially). This approach enables an improved parallel scaling, but limits the (reasonable) choice of the coupling frequency. To minimise errors, the coupling frequency should be chosen as small as possible, i.e., the smallest common multiple of the parent and the child model time step. For this reason a sensitivity analysis of different coupling frequencies is not provided here. We add this information to the new section "Model Performance".

**FIGURES: Figure 1 and 2 can be also smaller: I suggest to leave more space to enlarge figures as figure 4.**

We reduce the figure size for the revised version. However, the production office might do different things.

**Labels are not readable in figure 3a, 4, 10 and 11.**

Improved.

**Please uniform subpanels labelling (also add it to figure 10 and 11. I suggest to set white colour for near 0 values in figures 3a, 4 and 9.**

We added the subpanel labeling to Figs. 4, 7, 10 and 11.

We changed the color to white for near 0 values in Fig. 9 and to grey for Figs. 4 and 5 as the model domains should be distinguishable in the figures. However, for Fig. 3a there are no 0 values, as pink symbolises the EMAC domain.

Best regards,
Astrid Kerkweg and co-authors

Literature:

Hanke, M., Redler, R., Holfeld, T., and Yastremsky, M.: YAC 1.2.0: new aspects for coupling software in Earth system modelling, Geosci. Model Dev., 9, 2755-2769, doi:10.5194/gmd-9-2755-2016, 2016.

[Figure]

Figure 2: Vertical profiles of passive tracer (upper row) and their differences (lower row) for different topographic heights in the two COSMO/MESSy model instances (in $10^{-10}mol/mol$). The title gives the topographic height in the parent / child domain, respectively. Black line (triangles): initial profile in the parent model; blue line (circles): initial profile in the child model; red line (topdown triangles): coupled tracer profile in the parent model. The lower row displays the differences between the black and the red lines.

[Figure]

Figure 3: Time series of $SLP_{min}$ (upper row) and 10 m wind speed (lower row) (in the area of 10°N-50°N and 65°W-25°W) in the one-way (left) and 2-way (right) coupled simulation for EMAC (black), COSMO/MESSy$_{0.22°}$ (red), COSMO/MESSy$_{0.11°}$ (blue) based on 6-hourly data. The best-track intensity from HURDAT is shown as reference (green). EMAC is nuged until 29 September (dashed line, s. manuscript for details).

---

## Referee Comment (RC2) · Anonymous Referee #2 · 6 Aug 2017

This manuscript is part 5 of the documentation of the MECO(n) online coupled atmospheric chemistry system. It presents an update of the Multi-Model-DriverMMD from v1.0 to v2.0, which introduces the option of 2-way (as opposed to 1-way) coupling, and describes a new submodel GRID v1.0 for grid translations between the coupled model systems.

Online coupling of different model systems running on different grids, with different

input and output requirements, different time steps, etc. and running as separate executables is a challenging task. The authors have done a marvelous job in accomplishing such a coupling between a global and a regional model and even between different instances of the regional model. The work is obviously not yet complete, and thus this publication should be seen as another update or extension. The manuscript is composed of a main part and comprehensive user manuals of MMD and GRID as supplements. The supplements provide all the details that are important for users. While I did not have time to look at them in detail, they seem to be well organized and comprehensive with detailed information on data structures and routines and the overall logic. The main body of the manuscript is a high-level description of MMD and GRID and, in addition, presents a few example applications of the coupled system. To me this looks like an appropriate approach for a GMD publication.

General/major points: ———————— An important general question for a journal like GMD seems to me at what stage an update of a model deserves to be published. In my view only major updates that add significant new functionality should be published, and such updates should be in a mature stage. While the first criterion is clearly fulfilled by the present manuscript (a 2-way coupling is certainly a major update), the second point is much less clear, as detailed especially in point 1 of my main concerns below.

Although the manuscript is reasonably well structured and written and the topic is relevant and suitable for GMD, I have some major reservations: 1. The 2-way coupling is still in a pre-mature stage because a) dynamical 2-way coupling (of meteorology) seems not yet possible between EMAC and COSMO and even between instances of COSMO only seems to work reasonably well over flat terrain (an example is presented over the ocean). The main reason for this seems to be inconsistencies in the vertical grid transformations, especially because different methods are used for the partent-to-child (int2lm) and the child-to-parent (GRID v1.0) transformations. b) as just mentioned, the COSMO pre-processor tool int2lm is still used for the parent-to-child transformations, which seems redundant since GRID should provide all functionally required for

[Figure]

this, and using GRID for both up- and down-scaling would provide much more consistency. It remains unclear why this issue has not been resolved. c) The GRID submodel seems to be still in a fairly rudimentary stage. E.g. only "conservative remapping" is implemented (see P7, L6), whereas the description of the GRID submodel in Section 3 also mentions "interpolation" as a final goal (see P9, L15). Also vertical grid transformations seem to be implemented only in a pre-mature way (see next point).

2. Some parts of the manuscript are lacking clarity and detail. I am particularly missing details regarding the grid transformations and the separation into horizontal and vertical transfor-mations. In particular, COSMO is a non-hydrostatic model running on a geometrically fixed grid, whereas EMAC is hydrostatic and formulated on a hybrid pressure grid. Although I understand the motivation of the authors to keep the descriptions generic, the transformation between these fundamentally different vertical representations is essential and should be much better described. Furthermore, COSMO variables are represented on a staggered (Aralawa-C type) grid, which requires different transformations for variables like temperature or concentrations defined on grid cell centers, and variables like wind or tendencies defined on grid cell interfaces. Neither the main body of the manuscript nor the documentation makes any reference to the issue of staggered variables. The manuscript talks about a "geo-hybrid-grid" without explaining this structure and later about the "basegrid". GRID seems to expect a hybrid pressure grid (see line 14 on page 10), but how COSMO variables are transformed to hybrid pressure levels is never explained. Furthermore, if GRID only supports hybrid-pressure levels, it will be little suited for transformations between two instances of COSMO, as these are both operating on geometric grids. I am also missing information on details of the coupling, especially with respect to the fre-quency of the coupling: Are fields exchanged at every model time step? Are the parent and child models forced to use the same time steps? Is the frequency of coupling the same for the upward and the downward directions? Such information my be added in Section 2.2. 3. The authors emphasize the need for developing computationally efficient interfaces and submodels (e.g. line 19 on page 9), but no information is provided that would

allow the reader to judge the efficiency of the coupling that is ultimately achived. What is the computational overhead introduced by the coupling in terms of additional memory usage and computation time? Maybe this has been addressed in previous publications, but if so, this should be referenced. Otherwise, I would strongly encourage the authors to benchmark the model system (e.g. for one of the simulation examples in Sect. 4) with detailed timings of the individual model components and additional diagnostics, as this is a fundamental first step towards identifying bottlenecks and improving efficiency. The manuscript may be acceptable after addressing my main concerns 2 and 3 (plus the minor points below), or it may be postponed until a more mature version of coupling is available (i.e. main concern 1 is also addressed).

Minor points: ———— - Introduction: The reasons for the external coupling mentioned on P2/L20-30 are not entirely clear. Why is it good to "prevent the patches approach"? What are the "limitations of the Fortran95 namespace"? On the other hand, an advantage not mentioned is that this external coupling allows testing the influence of the coupling of different (individual) variables, which would likely be more difficult with internal coupling. The introduction should also emphasize the disadvantages and challenges of the external coupling, e.g. the challenge of transforming between different vertical grids.

- Footnote "2" on MPI-ESM seems little relevant in the context of this manuscript and could easily be deleted in my view.

- P3, L8: Sentence "This article documents the development of the . . .". No, this article is only part of a documentation.

- The following lines are presented in italics, which I found confusing until I realized that this is a citation. It would be clearer to present the references at the beginning and then the quoted text, e.g. "As described in Jöckel et al. (2015), Baumgaertner et al. (2016) and the Messy homepage (..), the Modular Earth System Model (MESSY) is "a sofware providing . . .".

Interactive
comment

- P4, L3: Delete the bracket "(Messyfied ECHAM ...)", this was already explained earlier.

- P4, L14: "update of MMD" -> "update of MMD presented here"

- At the end of the introduction section I was wondering whether GRID is now used for both directions replacing INT2LM entirely or not. It should already be explained here that the present implementation of GRID is only used for the child-to-parent transformation.

- P6, L15: What does "imprints its time settings" mean? Start and end of the simulation, time step, or something else? Does "imprint" mean that the child model has to use the same time step as the parent?

- P6, footnote 9: It would be better to include this information in the main text rather than as a footnote. Is it really necessary to distinguish between INT2LM and INT2COSMO in this manuscript?

- P7, L9-16: What is the difference between Option "0" and Option "1a"? On line 16, shouldn't it be Option "0" rather than "(a)", since (a) was introduced as an option available only for prognostic variables?

- P8, L17: The weight functions should remain the same during the simulation, at least the horizontal weights. Are the functions nevertheless transformed at each time step, i.e. the same transformation is repeated over and over again?

- P8, L28: The statement "for all required grid transformations" is not correct, since int2lm is used for partent-to-child transformations.

- P8, L29: I didn't understand this sentence. "Ideally" points at an ideal state not yet reached and should therefore be followed by "would be implemented" rather than "is implemented".

- P9, L1: What exactly do you mean by "as one central part of the model infrastructure"?
- P9, L5-8: I don't agree with the definition of regular and irregular grids. A "lambert conformal" grid as often used e.g. in WRF is also a regular orthogonal grid. A grid is usually regular in one projection but irregular (non-orthogonal) in another projection. Here it sounds like any non-lat-lon grid would be irregular (same issue in Section 3.1).

- Equally important as the horizontal grid transformation (and actually more challenging) is the vertical transformation. This needs more attention in section 3.

- P10, L11: What is a "geo-hybrid grid" structure?

- P10, L13: Why is a grid "defined by geographical longitude and latitude and vertically be hybrid pressure coefficients"? Is this a design choice for the GRID submodel? Does that imply that for a COSMO-COSMO nesting the COSMO grids (which may share the same projection) have to be first converted to geographical coordinates and then back to rotated ones? It would seem much more logical to me that GRID would translate everything to the same projection (e.g. the one used in the parent model), irrespective of whether it is a geographical coordinate system or not. How is the COSMO vertical grid transformed to hybrid pressures?

- Section 3.1.1 GRID_TRAFO: Grid transformations have been implemented in standard libraries like gdal (http://www.gdal.org/) and proj.4 (http://proj4.org), which also support rotated grids as used in COSMO. Why did you not choose to link to such a library that could provide a great level of flexibility? The SCRIP software seems to offer comparatively little flexibiliy. In my view one should strictly distinguish between coordinate translations (as can be accomplished by such libraries) and the final mapping between grids, which can be done by linear, cubic, or spline interpolation of any other (possibly conservative) mapping, and may be implemented as separate routines in GRID. Please make clear from the beginning that NREGRID is only implemented in GRID for vertical transformation, while SCRIP is used for all horizontal transformations, not only at the end of Section 3.1.3 (and more explicitly in the conclusions). Otherwise the reader - like myself - is confused about the role of NREGRID.

- P11, L1: Why is NREGRID recursive? Is this information relevant here? It sounds strange to me to have a recursive algorithm for grid translations.

- P12, L16: "For 2-way applications" -> "For 2-way coupling applications"

- P12, L21-26: A missing important point why size matters in atmospheric chemistry is that this chemistry is highly non-linear.

- P13, L7: dry deposition velocities do not only depend on soil type but also on turbulence, which could be another difference between the models.

- P15, L8: Is really only the pressure perturbation exchanged, i.e. the deviation from a reference pressure profile?

Typos and grammar: ———————— - P4, L30; "software as" -> "software such as"

- P5, L5: "reasonable" seems not the right word here.

- P7, L6: "At the time being" -> "For the time being"

- P7, L17: "For both option" -> "For both options"

- P12, L4: "handy" is not a good word in a scientific publication

- P12, L6: "tools, can" -> "tools can"

- P12, L31-32: Change to "If COSMO/Messy were 2-way coupled into EMAC and EMAC were using the NO emissions .."

- P13, L2: "what is mostly" -> "which is mostly"

- P13, L5: "pervious" -> "previous"

- P13, L6: I would say "slightly but systematically" rather than "systematically"

- P14, L4: "good" -> "well"

- P23, Fig. 5: "been mask" -> "been masked"

---

## Author Comment (AC2) · 26 Oct 2017

Note: Referee comments are indicated in black and bold, answers are in regular blue font, and changes to the revised manuscript are highlighted in lightblue.

**Dear Referee,**

thanks for the careful reading of our manuscript and valuable hints on shortcomings in its original form.

R1: This manuscript is part 5 of the documentation of the MECO(n) online coupled atmospheric chemistry system. It presents an update of the Multi-Model-DriverMMD from v1.0 to v2.0, which introduces the option of 2-way (as opposed to 1-way) coupling, and describes a new submodel GRID v1.0 for grid translations between the coupled model systems. Online coupling of different model systems running on different grids, with different input and output requirements, different time steps, etc. and running as separate executables is a challenging task. The authors have done a marvelous job in accomplishing such a coupling between a global and a regional model and even between different instances of the regional model. The work is obviously not yet complete, and thus this publication should be seen as another update or extension. The manuscript is composed of a main part and comprehensive user manuals of MMD and GRID as supplements. The supplements provide all the details that are important for users. While I did not have time to look at them in detail, they seem to be well organized and comprehensive with detailed information on data structures and routines and the overall logic. The main body of the manuscript is a high-level description of MMD and GRID and, in addition, presents a few example applications of the coupled system. To me this looks like an appropriate approach for a GMD publication.

**General/major points:**

An important general question for a journal like GMD seems to me at what stage an update of a model deserves to be published. In my view only major updates that add significant new functionality should be published, and such updates should be in a mature stage. While the first criterion is clearly fulfilled by the present manuscript (a 2-way coupling is certainly a major update), the second point is much less clear, as detailed especially in point 1 of my main concerns below.

A1: We do not share the referees concern about publishing the system in the current state for several reasons: First of all, major parts of the described 2-way coupling, in particular for atmosheric chemistry related applications and diagnostics are technically complete and already in use. These applications deserve a proper reference, which can be cited if it comes to publications. Second, the current development step was released with the release 2.53 of our model system (see Section "Code

Availability") to a wider user community. This requires a proper reference and documentation, including a clear description of the current limitations.

- R2: Although the manuscript is reasonably well structured and written and the topic is relevant and suitable for GMD, I have some major reservations: 1. The 2-way coupling is still in a pre-mature stage
- A2: As above, we disagree. The 2-way coupling is technically complete and the updated model system is already in use, in particular for atmospheric chemistry related applications. These are in the focus of our work. Thus, we think, publication of the current stage of the development is justified, despite the fact that the dynamical 2-way coupling, which is a specific challenge on its own, requires further development.
- R3: because a) dynamical 2-way coupling (of meteorology) seems not yet possible between EMAC and COSMO and even between instances of COSMO only seems to work reasonably well over flat terrain (an example is presented over the ocean). The main reason for this seems to be inconsistencies in the vertical grid transformations, especially because different methods are used for the partent-to-child (int2lm) and the child-to-parent (GRID v1.0) transformations.
- A3: This is correct for the EMAC-COSMO coupling. However, for the COSMO-COSMO coupling the vertical interpolation routine of INT2LM is used for both directions. We have to admit that this is not clear in the paper, but well documented in the supplement.

We add this information to the revised manuscript. The major problem for both, EMAC-COSMO and COSMO-COSMO couplings, is that INT2LM does not only perform a vertical remapping, but makes a crude assumption of keeping the boundary layer (i.e. up to 850 hPa) as it is, however, moving it to the height of the target orography and remapping only the remaining part of the vertical column. This procedure, as it is implemented in INT2LM, is not reversible and thus introduces spurious effects for different orographies, which are always present because of the different horizontal resolutions of the nested models.

- R4: b) as just mentioned, the COSMO pre-processor tool int2lm is still used for the parent-to-child transformations, which seems redundant since GRID should provide all functionality required for this, and using GRID for both up- and down-scaling would provide much more consistency. It remains unclear why this issue has not been resolved.
- A4: INT2LM provides much more functionalities than only remapping. It reads and processes the external data required as input for the COSMO model (especially for the initialisation of the model). Moreover, it performs some field adjustments concerning inconsistencies between the land-sea-mask of the COSMO model and the in-coming data.

We add this information to the introduction of Sect. 2.

Therefore, it is not possible to completely eliminate INT2LM. One could, however, indeed exchange the horizontal and vertical interpolation routines. We started to test this, but in the first place the performance (w.r.t. the results) of the child model was downgraded. As explained above, the main problem is the extra treatment of the boundary layer, which for the off-line nested COSMO model is the preferred way, as this makes physically more sense compared to a simple vertical

interpolation. But in terms of reversibility of the height adjustment procedure this unfortunately causes problems.

- R5: c) The GRID submodel seems to be still in a fairly rudimentary stage. E.g. only "conservative remapping" is implemented (see P7, L6), whereas the description of the GRID submodel in Section 3 also mentions "interpolation" as a final goal (see P9, L15). Also vertical grid transformations seem to be implemented only in a pre-mature way (see next point).
- A5: This is obviously a misunderstanding that we need to clarify. Please do not mix up GRID with the applied interpolation procedure for the 2-way coupling in MMD2WAY. GRID is fully functional, it includes two remapping packages: NREGRID and SCRIP, implying that all interpolation / remapping routines provided by NRE-GRID and SCRIP can be used with the GRID submodel.

For the 2-way coupling (MMD2WAY), however, only conservative remapping is utilized so far, for mainly two reasons: First, an improved run-time performance can be achieved by efficiently exploiting the given horizonal domain decompositions of the models, because the data exchange between the local domains could be minimised in comparison to other remapping or interpolation schemes. And second, conservative remapping is the first choice for atmospheric chemistry applications, since fluxes (e.g. emission fluxes) are conserved.

We are more precise about this in the revised manuscript.

We change the sentence on page 7 from "For the time being, only conservative remapping is implemented as horizontal transformation method" to "For the time being, only the conservative remapping, as provided by GRID, is utilized as transformation method in MMD2WAY\_CHILD."

- R6: Some parts of the manuscript are lacking clarity and detail. I am particularly missing details regarding the grid transformations and the separation into horizontal and vertical transformations. In particular, COSMO is a non-hydrostatic model running on a geometrically fixed grid, whereas EMAC is hydrostatic and formulated on a hybrid-pressure grid. Although I understand the motivation of the authors to keep the descriptions generic, the transformation between these fundamentally different vertical representations is essential and should be much better described.
- A6: The revised manuscript contains a more detailed description.
- R7: Furthermore, COSMO variables are represented on a staggered (Arakawa-C type) grid, which requires different transformations for variables like temperature or concentrations defined on grid cell centers, and variables like wind or tendencies defined on grid cell interfaces. Neither the main body of the manuscript nor the documentation makes any reference to the issue of staggered variables.
- A7: The only variables defined on the staggered grid in the COSMO model are the wind components. For the COSMO-EMAC coupling the COSMO wind components are first interpolated to the cell midpoints, and afterwards transformed to the EMAC grid. For the COSMO-COSMO coupling the wind components are interpolated directly between the staggered grids, i.e., they are always defined on the box edges. This information is added to the revised manuscript and the MMD User Manual.
- R8: The manuscript talks about a "geo-hybrid-grid" without explaining this structure [...]

A8: The Fortran data structures of GRID are based on those of NCREGRID as published by [Jöckel(2006)]. This article provides an extensive explanation of the definition of a geo-hybrid grid. Therefore, we did not repeat it in the current article.

[Jöckel(2006)] introduces the geo-hybrid grid as follows: The horizontal grid space of a geo-hybrid grid usually comprises geographical latitude and longitude. Especially in 3-D global atmospheric models the vertical pressure (p) coordinate is often defined by hybrid levels (with index i) of the form

$$p(i, x, y, t) = h_a(i) \cdot p_0 + h_b(i) \cdot p_s(x, y, t),$$
(1)

where  $p_s$  is the surface pressure,  $p_0$  is a constant reference pressure, and  $h_a$  and  $h_b$  are the dimensionless hybrid coefficients. This representation in a curvi-linear coordinate system (dependent on longitude x, latitude y, and time t) allows a terrain following vertical coordinate, if  $h_a = 0$  and  $h_b = 1$  for the lowest level (surface level)."

Thus, dependent on the choice of  $p_s$  and  $p_0$  GRID is capable to handle all cases of vertical pressure axes, such as

- hybrid pressure axes  $(h_a \neq 0, h_b \neq 0)$ ,
- constant pressure axes  $(h_b = 0)$  and
- sigma levels  $(h_a = 0)$ .

In the case of  $h_b = 0$ ,  $h_a \cdot p_0$  could also be a height coordinate. The grid transformations require that source and destination grids are both defined in the same vertical representation, i.e. either with pressure or with height coordinates.

We add some more information to the revised Sect. 3.1 (which also required a slight reordering), with a specific reference to the corresponding section in the revised GRID User Manual, which contains the complete information.

- R9: [...] and later about the "basegrid".
- A9: "basegrid" is a short cut for one specific geo-hybgrid grid, namely the 3-D grid of the basemodel (see Sect. 3.2 in the GMDD version of the manuscript).
- R10: GRID seems to expect a hybrid pressure grid (see line 14 on page 10), but how COSMO variables are transformed to hybrid pressure levels is never explained.
- A10: The COSMO model still provides the possibility to define the vertical grid by pressure levels. This option was still frequently used when our model development started. Thus, for the definition of the hybrid pressure grid, we currently use the routines provided by INT2LM. Nevertheless, in the meantime GRID is further developed to deal with the requirements of the ICON model, which only features height axes. Thus, we are going to implement the possibility to use the actual (time-dependent) 3-D pressure field for remapping between height and pressure grids, since this will be required for the MESSy infrastructure submodel IMPORT connected to ICON.

We discuss this issue more in the revised Sect. 2.2.

R11: Furthermore, if GRID only supports hybrid-pressure levels, it will be little suited for transformations between two instances of COSMO, as these are both operating on geometric grids.

- A11: As explained in answer A8, height height interpolations are possible as well. Anyhow, as stated in answer A3, for the back-transition between two COSMO instances the procedure of INT2LM is used.
- R12: I am also missing information on details of the coupling, especially with respect to the frequency of the coupling: Are fields exchanged at every model time step? Are the parent and child models forced to use the same time steps? Is the frequency of coupling the same for the upward and the downward directions? Such information my be added in Section 2.2. 3.
- A12: Parent and child model do not need to use the same time step lengths. However, the parent model time step length needs to be a common multiple of the child model time step length. The coupling frequency can be changed in the namelist, but to minimise the deviation of the child model from the parent model state at the boundary, it is strongly recommended to couple every parent model timestep. This information is provided in a new subsection "Model performance" of the revised manuscript.
- R13: The authors emphasize the need for developing computationally efficient interfaces and submodels (e.g. line 19 on page 9), but no information is provided that would allow the reader to judge the efficiency of the coupling that is ultimately achieved. What is the computational overhead introduced by the coupling in terms of additional memory usage and computation time? Maybe this has been addressed in previous publications, but if so, this should be referenced. Otherwise, I would strongly encourage the authors to benchmark the model system (e.g. for one of the simulation examples in Sect. 4) with detailed timings of the individual model components and additional diagnostics, as this is a fundamental first step towards identifying bottlenecks and improving efficiency.
- A13: It is not clear to us, to which reference this "overhead" and additional computation time should be compared to? An off-line 2-way nesting is hardly possible, and the show-stopper would obviously be the tremendous I/O required to write and read the files with coupled fields in every model time step. The required memory (for the 2-way nesting) increases linearly with the number of variables that need to be exchanged. And the run-time performance depends first and foremost on the specific model setups (e.g., on the complexity of the chosen chemistry representation etc.). But most important, the overall performance is at the end determined by the "degree of balance" of the distribution of parallel tasks among the different model instances. We discussed this in detail in Part II of our series (Kerkweg and Jöckel, 2012) for the 1-way nesting case. The same principles hold for the 2-way exchange, except for the complication that communication waiting times depend now on bidirectional data exchange. Thus, it is up to the user to find (experimentally) the optimum task distribution to minimise communication waiting times.

We add a brief discussion on this to the new Section "Model performance" and additionally assess the dependency of the simulation time on the number of coupled fields in order to check, whether an increasing number of fields shows discontinuously prolonged simulation times caused by the 2-way exchange.

R14: The manuscript may be acceptable after addressing my main concerns 2 and 3 (plus the minor points below), or it may be postponed until a

more mature version of coupling is available (i.e. main concern 1 is also addressed).

- A14: See our answers A1 and A2.
- R15: Minor points: Introduction: The reasons for the external coupling mentioned on P2/L20-30 are not entirely clear. Why is it good to "prevent the patches approach"? What are the "limitations of the Fortran95 namespace"?
- A15: Maybe we were not precise enough here. The "patches approach" is usually a feature of regional grid-refinements, which is directly embedded in (or part of) the model code, as for instance in WRF or ICON, in which the user can specify the number of patches and their corresponding domains flexibly at run-time. For such a feature, however, the entire model code needs to be "aware" of a(n arbitrary) number of grid-refined patches. Thus, this needs to be implemented "by design". To equip legacy code (as COSMO or ECHAM) supplementarily with such a feature would basically mean to rewrite the entire code from scratch. The reason is that all prognostic (and diagnostic) variables need to exist on each patch (technically independent of each other). How this is technically achieved depends largely on the applied programming language. In fully object oriented languages, overloaded "sets" or "instances" of the same variable(s) could be defined, however, the Fortran95 language standard does not allow to have the same variable with the same name in the same name-space more than once. Thus a complete recoding, e.g., replacing arrays by structures of arrays is required.

The first /second part of this answer is added for clarification to the first / second bullet point in the introduction of the revised manuscript.

R16: On the other hand, an advantage not mentioned is that this external coupling allows testing the influence of the coupling of different (individual) variables, which would likely be more difficult with internal coupling. The introduction should also emphasize the disadvantages and challenges of the external coupling, e.g. the challenge of transforming between different vertical grids.

A16: First of all, thank you very much for this important hint on variable testing. We add this point to our revised list of advantages.

Indeed, in our current applications we do exactly this (e.g., chemical 2-way nesting with dynamical 1-way nesting).

Concering the challenges: The need to transform between different vertical grids is not necessarily connected to the way of (internal or external) coupling. Nevertheless, the patch (or grid-refinement) approaches are usually implemented as "internal" coupling and do keep the vertical grid between different patches in order to avoid vertical interpolation. But also in an external coupling approach the vertical grid between different model instances can be the same. In both cases, however, the issues due to the horizontally refined orography information remain.

To expand the discussion, we add some statements about the disadvantages and challenges of external coupling to the revised introduction, right after the list of reasons for choosing external coupling:

"Apart from these advantages, the external coupling proves to be more challenging than the internal coupling. Horizontal and vertical interpolation errors are expected to be larger, depending on the relations between the different grids and differences in the orography. From these, the adaption to the higher resolved orography of the nested simulation causes the largest error. An additional disadvantage of all external coupling approaches is the need for the user to optimise the distribution of the available parallel tasks among the different model instances, in order to achieve an optimal run-time performance with minimized waiting times between the model instances."

- R17: Footnote "2" on MPI-ESM seems little relevant in the context of this manuscript and could easily be deleted in my view.
- A17: We want to provide references for each model. Instead of the web-site, we now cite Giorgetta et al. (2013).
- R18: P3, L8: Sentence "This article documents the development of the . . .". No, this article is only part of a documentation.
- A18: We change the statement to "This article documents a major achievement in the development of the ... "
- R19: The following lines are presented in italics, which I found confusing until I realized that this is a citation. It would be clearer to present the references at the beginning and then the quoted text, e.g. "As described in Jöckel et al. (2015), Baumgaertner et al. (2016) and the MESSy homepage (..), the Modular Earth System Model (MESSy) is "a sofware providing . . .".
- A19: Thanks for this suggestion! Indeed, the reordering enhances the readability a lot. Changed.
- R20: P4, L3: Delete the bracket "(Messyfied ECHAM . . .)", this was already explained earlier.
- A20: Done.
- R21: P4, L14: "update of MMD"  $\rightarrow$  "update of MMD presented here"
- A21: Changed.
- R22: At the end of the introduction section I was wondering whether GRID is now used for both directions replacing INT2LM entirely or not. It should already be explained here that the present implementation of GRID is only used for the child-to-parent transformation.
- A22: We change the sentence "Sect. 3 introduces the newly developed GRID submodel, which provides the required mapping functionalities" to "Sect. 3 introduces the newly developed GRID submodel, which provides the required mapping functionalities used for the child-to-parent data exchange.". Furthermore, the last sentence of the prior paragraph reads now "GRID can be used for all grid mapping operations required during a simulation." instead of "GRID is used for all grid mapping operations required during a simulation."
- R23: P6, L15: What does "imprints its time settings" mean? Start and end of the simulation, time step, or something else?
- A23: These are the start-date (only if a model instance is newly started), the end-date, and the restart trigger.Item changed to "the parent imprints its time settings on the child model, i.e., enddate, restart trigger and, at the very first start of a model instance, the (re-)start-date as start-date of this instance."

- R24: Does "imprint" mean that the child model has to use the same time step as the parent?
- A24: No. Forcing the coarsest instance to use the same short time step length as the finest resolved model instance would downgrade the performance of the system dramatically. Nevertheless, the time step lengths of all model instances need to be common multiples.

For respective changes in manuscript: see A12.

- R25: P6, footnote 9: It would be better to include this information in the main text rather than as a footnote. Is it really necessary to distinguish between INT2LM and INT2COSMO in this manuscript?
- A25: We prefer to keep the differentiation between INT2LM and INT2COSMO to keep the manuscript consistent with Part II of the article series about MECO(n). We inline the footnote in the revised manuscript. The sentence reads: "Afterwards the data is transformed from the in-grid to the child grid using the expanded version of the preprocessing software INT2LM for the COSMO model (INT2COSMO). See Kerkweg and Jöckel (2012b) for further explanations."
- R26: P7, L9-16: What is the difference between Option "0" and Option "1a"? On line 16, shouldn't it be Option "0" rather than "(a)", since (a) was introduced as an option available only for prognostic variables?
- A26: With option "0", as explained in the text, the memory for the target field is allocated within MMD2WAY\_PARENT and can afterwards be accessed by other MESSy submodels. For option "1", however, the variable needs to exist in the parent model and will be modified directly.

In the revision, we change the wording from "the field is used to modify an existing parent model field." to "the exchanged field is used to directly modify a parent model field." Additionally, as the explanations below belong all to option "1", we moved the end of the enumeration to the end of the section.

- R27: P8, L17: The weight functions should remain the same during the simulation, at least the horizontal weights. Are the functions nevertheless transformed at each time step, i.e. the same transformation is repeated over and over again?
- A27: Indeed, our sentence is misleading. We change it to:"They are once, during the initialization phase, transformed in the same way as the data and sent to the parent model for application during the integration phase."
- R28: P8, L28: The statement "for all required grid transformations" is not correct, since int2lm is used for partent-to-child transformations.
- A28: This needs indeed to be clarified. GRID is independent of MMD2WAY. The MESSy infrastructure component GRID provides the basis for all required grid transformations. However, in the MMD2WAY\_CHILD submodel we decided to use the INT2LM software instead of GRID. One of the reasons is that the 1-way online coupling was implemented prior to GRID.

This sentence is skipped in the revised manuscript anyhow, see A29.

R29: P8, L29: I didn't understand this sentence. "Ideally" points at an ideal state not yet reached and should therefore be followed by "would be implemented" rather than "is implemented".

- A29: Yes. "Ideally" means that in the best case, the GRID submodel provides all the listed functionalities. So far, not all of them are implemented. This is clarified in the revised article. We change "is" to "would be". Moreover, we see that the first sentence of this subsection is misleading. Therefore we skip it and add a more general introduction, fitting better the following more general statements. The new sentence reads: "Due to the increasing complexity of Earth System Models, grid transformations at run-time of the model, (e.g., remapping from an atmosphere grid to a higher resolved land grid and vice versa) are more and more commonly required. To avoid individual implementations throughout the code, such an on-line transformation functionality should be implemented as one important part of the model infrastructure, providing a common grid processing functionality."
- R30: **P9**, **L1**: What exactly do you mean by "as one central part of the model infrastructure"?
- A30: This seems to be a misunderstanding due to incorrect use of language. Grid transformation is only one of a number of important parts in a model infrastructure. Others are, for instance, memory management or time and event handling. Therefore we rephrase to "one important part".
- R31: P9, L5-8: I don't agree with the definition of regular and irregular grids. A "lambert conformal" grid as often used e.g. in WRF is also a regular orthogonal grid. A grid is usually regular in one projection but irregular (non-orthogonal) in another projection. Here it sounds like any non-latlon grid would be irregular (same issue in Section 3.1).
- A31: Indeed, it was definitely not our intention to name every non-lat-lon grid irregular! Following the grid classifications of Bowler and Clegg (2011), we decided to change the grid classification to
  - rectangular grids, which are either orthogonal in geo-coordinates (rectilinear grids) or curvi-linear grids,
  - non-rectangular structured grids, and
  - unstructured grids.

The second sentence of Sect. 3.1 is changed accordingly to "Four different grid types are distinguishable: (1) rectangular grids, which are orthogonal in geo-coordinates, (2) curvi-linear grids, (3) non-rectangular, structured grids and (4) unstructured or irregularly geo-located grids.

- R32: Equally important as the horizontal grid transformation (and actually more challenging) is the vertical transformation. This needs more attention in section 3.
- A32: We provide additional information on the vertical remapping in revised Sect. 3. by introducing a new subsection "3.1.4 Application of GRID in MMD2WAY\_CHILD".
- R33: P10, L11: What is a "geo-hybrid grid" structure?
- A33: A geo-hybrid grid structure is the Fortran type definition (= structure), which contains all data required to define a geo-hybrid grid (for the definition of a geo-hybrid grid see answer A8.)

We change the sentence to "The definition of the Fortran structure, which contains

all components required for the definition of a geo-hybrid grid, was extended and generalised for the usage in GRID."

- R34: P10, L13: Why is a grid "defined by geographical longitude and latitude and vertically by hybrid pressure coefficients"? Is this a design choice for the GRID submodel?
- A34: Yes it is. See answer A8.
- R35: Does that imply that for a COSMO-COSMO nesting the COSMO grids (which may share the same projection) have to be first converted to geographical coordinates and then back to rotated ones? It would seem much more logical to me that GRID would translate everything to the same projection (e.g. the one used in the parent model), irrespective of whether it is a geographical coordinate system or not.
- A35: The referee comments R35, R37 and R39 point to an additional misunderstanding: In our article the term "grid transformations" refers always to data "remapping", "regridding" or interpolation between different model grids based on geographical coordinates. We never intended to use it in the meaning of "grid translations" or "map projection".

Independent of the computational grid of the model, the grid structure in GRID contains the data of the grid vertices and / or cell centers always in geographical coordinates. This information is – in all cases – provided by the respective base-models (ECHAM5 and COSMO) and does not need to be determined by the GRID submodel. Thus, GRID does not need to perform "grid translations" or "map projections". The remapping weights of geo-located data between two different grids are always calculated in geographical coordinates.

We change the term transformation to remapping at some location in the article to avoid the above misunderstanding. Additionally, the information about the geographical coordinates is added to the revised Sect. 3.1.

- R36: How is the COSMO vertical grid transformed to hybrid pressures?
- A36: Until now, the hybrid pressure coefficients calculated by INT2LM or the COSMO model are used. See answer A10.
- R37: Section 3.1.1 GRID\_TRAFO: Grid transformations have been implemented in standard libraries like gdal (http://www.gdal.org/) and proj.4 (http://proj4.org), which also support rotated grids as used in COSMO. Why did you not choose to link to such a library that could provide a great level of flexibility? The SCRIP software seems to offer comparatively little flexibility. In my view one should strictly distinguish between coordinate translations (as can be accomplished by such libraries) and the final mapping between grids, which can be done by linear, cubic, or spline interpolation of any other (possibly conservative) mapping, and may be implemented as separate routines in GRID.
- A37: As far as we understand the functionalities of these libraries, they would not help, as it is not coordinate translations (or map projections) we need, but the actual remapping of data between different geographical coordinate systems.
- R38: Please make clear from the beginning that NREGRID is only implemented in GRID for vertical transformation, while SCRIP is used for

all horizontal transformations, not only at the end of Section 3.1.3 (and more explicitly in the conclusions). Otherwise the reader - like myself - is confused about the role of NREGRID.

A38: Sorry! Obviously we have to be more clear about the separation of GRID and the MMD submodels. NREGRID was originally implemented for horizontal and 3-D spatial remapping. This is still the standard way for the data import in the EMAC model (i.e., with ECHAM5 as basemodel).

In the COSMO model, some of the requirements of NREGRID w.r.t. the grid structure are, however, not fullfilled. Therefore, we had to introduce a second remapping option for horizontal grids, which can deal with rotated grids, such as the COSMO grid. We decided to use the very well known and commonly used SCRIP software package.

Therefore, MMD2WAY\_CHILD uses SCRIP for the horizontal interpolations, and NREGRID for the vertical remapping, as SCRIP does not provide remapping along the vertical axis.

- R39: P11, L1: Why is NREGRID recursive? Is this information relevant here? It sounds strange to me to have a recursive algorithm for grid translations.
- A39: NREGRID is not for grid translations. It is for the rediscretisation of "gridded" geo-scientific data between n-dimensional (usually n = 2 or 3) orthogonal grids. The conservative rediscretisation of extensive or intensive variables is based on the calculation of the overlap (area or volume) matrix between source and destination grid boxes. For orthogonal grids these overlap matrices can nicely be calculated recursively, since the overlap area / volume is zero as soon as at least the overlap interval along one axis (dimension) is zero. For details see Jöckel (2006). Since the recursive nature of this algorithm limits its application to orthogonal grids it cannot be applied for rediscretisations between the (in geographical coordinates) orthogonal Gaussian grid of ECHAM5 and the rotated (in geographical coordinates non-orthogonal) COSMO grid. This is why we needed to implement SCRIP as well. Thus, the information is relevant.

We add this information to Sect. 3.1.2.

R40: P12, L16: "For 2-way applications"  $\rightarrow$  "For 2-way coupling applications" A40: Changed.

R41: P12, L21-26: A missing important point why size matters in atmospheric chemistry is that this chemistry is highly non-linear.

A41: This is indeed what we meant. We rephrase the last sentence to "Especially in highly polluted regions, or more generally near emission sources, this might influence the simulated chemical regime, as atmospheric chemistry is highly non-linear."

- R42: P13, L7: dry deposition velocities do not only depend on soil type but also on turbulence, which could be another difference between the models.
- A42: Thanks for this remark.

We change the last sentence from ", which is most propably due to different soil properties in that region." to ", which is most propably due to different soil properties and also due to the different turbulence schemes employed by the two basemodels. "

**R43: P15, L8: Is really only the pressure perturbation exchanged, i.e. the deviation from a reference pressure profile?**

A43: For the COSMO-COSMO coupling only the deviation of the pressure from the reference atmosphere is exchanged during the integration. During the model initialisation phase all information required for the definition of the parent grid are exchanged. The definition of the reference atmosphere itself is part of this onetime data exchange. The term "pressure perturbation" seems to lead to a misunderstanding.

Therefore we change "pressure perturbation (pp)" by "the pressure deviation from the reference atmosphere (PP)".

**Typos and grammar:**

- P4, L30; "software as"  $\rightarrow$  "software such as" Corrected.

**P5**, **L5**: "reasonable" seems not the right word here. "This is reasonable," is replaced by "This was required,"

P7, L6: "At the time being"  $\rightarrow$  "For the time being" Corrected.

P7, L17: "For both option"  $\rightarrow$  "For both options" Corrected.

P12, L4: "handy" is not a good word in a scientific publication Replaced by "useful".

**P12, L6:** "tools, can"  $\rightarrow$  "tools can" Corrected.

P12, L31-32: Change to "If COSMO/Messy were 2-way coupled into EMAC and EMAC were using the NO emissions ..."

P13, L2: "what is mostly"  $\rightarrow$  "which is mostly" Corrected.

P13, L5: "pervious"  $\rightarrow$  "previous" Corrected.

P13, L6: I would say "slightly but systematically" rather than "systematically"

Done.

P14, L4: "good"  $\rightarrow$  "well" Corrected.

**P23, Fig. 5:** "been mask"  $\rightarrow$  "been masked" Corrected.

Best regards, Astrid Kerkweg and co-authors

**References**

- [Blower and Clegg(2011)] Blower, J. and Clegg, A.: Fast regridding of large, complex geospatial datasets, in: Com.Geo 2011: The 2nd International Conference on Computing for Geospatial Research & Applications, pp. 1-6, URL http://centaur.reading.ac.uk/19928/, 2011.
- [Giorgetta et al.(2013)] Giorgetta, M. A., Jungclaus, J., Reick, C. H., Legutke, S., Bader, J., Böttinger, M., Brovkin, V., Crueger, T., Esch, M., Fieg, K., Glushak, K., Gayler, V., Haak, H., Hollweg, H.-D., Ilyina, T., Kinne, S., Kornblueh, L., Matei, D., Mauritsen, T., Mikolajewicz, U., Mueller, W., Notz, D., Pithan, F., Raddatz, T., Rast, S., Redler, R., Roeckner, E., Schmidt, H., Schnur, R., Segschneider, J., Six, K. D., Stockhause, M., Timmreck, C., Wegner, J., Widmann, H., Wieners, K.-H., Claussen, M., Marotzke, J., and Stevens, B.: Climate and carbon cycle changes from 1850 to 2100 in MPI-ESM simulations for the Coupled Model Intercomparison Project phase 5, Journal of Advances in Modeling Earth Systems, 5, 572–597, doi:10.1002/jame.20038, URL http://dx.doi.org/10.1002/jame.20038, 2013.
- [Jöckel(2006)] Jöckel, P.: Technical note: Recursive rediscretisation of geo-scientific data in the Modular Earth Submodel System (MESSy), Atmos. Chem. Phys., 6, 3557–3562, 2006.
- [Kerkweg and Jöckel(2012b)] Kerkweg, A. and Jöckel, P.: The 1-way on-line coupled atmospheric chemistry model system MECO(n) - Part 2: On-line coupling with the Multi-Model-Driver (MMD), Geoscientific Model Development, 5, 111-128, doi: 10.5194/gmd-5-111-2012, URL http://www.geosci-model-dev.net/5/111/2012/, 2012b.

---

## Author Response (AR2)

Dear Sophie,

thank you very much for the in-depth reading of our manuscript and the valuable additional comments!

Please find below the detailed answers to your comments including the resulting changes made to our re-revised manuscript, (appended with highlighted changes) and the new supplemental material. Please note that

- on page 3, line 28 of the articles latexdiff, only the "last access date" of the website was changed from "04 October 2017" to "15 November 2017".

- on page 18, the link to the IBTracs data was no longer available. Instead, a reference to the respective DOI was added.

It is a problem of latexdiff, that these differences do not show up correctly.

**Dear author,**

**Thank you very much for your revised manuscript and the details with which you answer the two referees' concerns. I consider you globally satisfied their remarks besides few points that I list here and this is why I am asking for an "editor review". So thank you for addressing the issues below that I think are important before considering publication of the manuscript. (Please note that when I cite pages and lines, I refer to your gmd-2017-87-author_response-version3.pdf document).**

**Major comments**

**1) English : I suspect that the text has not been written by an english native writer. In many places the text is heavy and not easy to read. Please have the english revised by an english native speaker. For example:**
**-The sentence "The question, if the aggregation of the subgrid-scale information provided by the smaller scale model to the larger scale model constitutes an added value for the larger scale model is still under debate and might be answered with the help of 2-way coupled applications." could be changed for "The fact that whether or not the aggregation ... is still under debate and this question might be answered with the help of 2-way coupled applications"**
**-p. 19, the sentence "the two slices of the boundary fields, for which COSMO performs a linear time interpolation, are filled with the data of the actual time step" could be changed for " the two time slices of the boundary fields between which COSMO usually performs a linear time interpolation are now filled with the data of the actual time step**

You are right. None of the authors is a native English speaker. Unfortunately, no native English speaker is at hand at the institutes. Nevertheless, we changed the two sentences you refer to and improved the writing of the language at some other places. As the articles are anyhow copy edited by Copernicus in the final production process, a final quality check is anyway on the way.

**2) Regarding the section on model performances (comments c by referee #1 and R13 by referee #2 R13), you added a new section, but only the computing time for data transformation is shown while a more interesting measure**

**would be the elapse time of the whole coupled system and an analysis on how the two-way coupling impacts this balance with respect to the one-way coupling. Is it possible to have few numbers on comparative elapse time and load balance of the two-way coupled system wrt the one-way coupled system?**

It is difficult to get reliable numbers here due to several facts. The first is a purely technical issue: currently, the MECO(n) system can not be analysed using the Aliena-software, which is the software available for program analyses at DKRZ, where we perform our model developments and simulations.

Secondly, the run-time performance depends very much on the chosen setup. The largest impacts are

- the complexity of the simulation itself (e.g., whether or not chemistry is included),

- the number of coupled fields,

- the sizes of the coupled domains,

- the number of chosen tasks, and

- the task distribution between the different model instances.

Thus, all numbers that might be shown here, are only valid for exactly the one specific setup as analysed in the example. We apprehend that these numbers might be misleading and could be wrongly interpreted. Therefore we are hesitating to show such numbers in the main body of the article. However, as a compromise, we include the analysis given below in the supplement, and add a reference to it in Sect. 5.

Using the run time information provided by the scheduler, we can provide very rough numbers:

Table 1 shows the run time in node hours for a 1 day simulation of one specific MECO(2) setup, applied with four different task distributions (S01-S04) on 3 nodes (à 24 tasks) at the Mistral computer at Deutsches Klimarechenzentrum (DKRZ). For the mere dynamical setups S02 is the fastest, as could be expected from the number of horizontal grid boxes in CM1 and CM2 and the fact that only the child model has to perform the data transformations. This is still the case for the 1-way simulations including 139 tracers. However, the additional time required for the 2-way coupling is largest in the case with 139 tracers for this setup (45.3%). This shows that the required additional time for the 2-way nesting is extremly setup dependent. Thus, the best task distribution needs to be chosen by the user for each setup individually. Here, the user has to take into account at least the size of the different model domains, the number of 1-way and / or 2-way coupled variables, and the complexity of the simulation, e.g. whether or not chemistry is included.

On the first glance the required additional time seems to be large. However, the tracers in this example are only transported. No chemical kinetic calculations are included. Including these, at least 10 times higher run times are to be expected. Thus the additional run time for the 2-way nesting would shrink dramatically relative to the overall run time of chemical applications.

**3) Following comments by referee #1 and #2 on the meaning of "Ideally" in section 3, I consider that your introduction on grid transformation algorithm is way too general and too long. This whole section should be revised to clearly describe on what GRID really provides and not on what an ideal**

library would provide.

We adapted the beginning of the section according to your suggestion.

**One specific comment: given your description of grids, I don't understand what is the difference between "curvi-linear" and "non-rectangular strcutured" grids; can you be more precise or give an example (but only if if these grids are relevant for GRID).**

GRID v1.0 deals with rectangular grids only. A curvi-linear grid is e.g., a rotated regional domain as usually used for COSMO or WRF. A non-rectangular, structured grid is, e.g., the icosahedral grid used by the ICON model. We added this information to the article.

**Finally, the second sentence of first paragraph of 3.1 "Three different grid types ..." is completely redundant with the introduction of section 3 (lines 20-25)**

This changed in consequence of the adaptions listed above.

**Minor comments:**

**1) The modifications done on what is now p.8 regarding the use of "0" and "1" should also be done on what is now p.9 for the use of "0", "1" and "2"**

Unfortunately, we are not sure which modifications you are refering to. Thus we only added quotation marks.

**2) Regarding your Figure 2, you write that you want to add it to the supplement and make a reference to the supplement in the text. First, I don't see where that reference is done. Second, on the figure, I don't see any black line (triangles) so I think it is not appropriate to refer to these in the captions.**

Sorry! Somehow we missed to include it. The reference to the supplement and Fig.2 is now added to the end of Sect. 3.1.

**3) I think that "optimisation" in "without any resolution dependent optimisation of the emission scheme" could be changed for "tuning".**

Changed.

**4) R3 and A3: Regarding the vertical grid transformations, I think your text is more precise now but you still should mention the inconsistencies linked to the different methods used for the parent-to-child and for the child-to-parent vertical transformations**
**5) R4 and A4: Regarding the use of INT2LM, I think you should clarify in the text why you did not exchange the horizontal and vertical interpolation routines as you explain in your author response.**

To comply with your comments 4+5, we added a paragraph at the end of Sect. 3.1.

The beginning of Sect. 4 was changed accordingly to avoid repeatings.

**6) R15: I still don't understand what is the "patch approach". You write: "The user can specify the number of patches" but patches of what? Please explain better as it is not evident at all for non regional modellers.**

The term "patch" is usually used for the different grid refinement areas in a model providing the on-line nesting capability. Each refinement instance is called a "patch". For example, the german weather prediction model ICON is currently operated with two patches: the global patch (at 13 km) and the regional patch (ICON-EU, at 6.5 km) covering most of Europe. (for an illustration see the first picture on

https://www.dwd.de/DE/forschung/wettervorhersage/num_modellierung/01_num_vorhersagemodelle/icon_beschreibung.html?nn=512942,

last access date: 14.11.2017). Technically, the important point is, that the refinement possibility is part of the model code itself. In contrast to this, the COSMO model can only be operated at one resolution at a time. For the off-line nesting of finer resolved instances, an additional preprocessing software, as INT2LM in the case of COSMO, needs to be applied to drive this finer resolved model instance. We changed the text when introducing the term "patch".

**7) p.5, you write "GRID can be used for all grid mapping operations ..."; I suppose this is not true as in practice it cannot be used for data going into COSMO.**

It is true. It can be used for all grid mapping applications, but in our case it is not used. This is because, (1) the 1-way nesting was implemented prior to GRID (see answer A28 in reply to referee # 2), and (2), unfortunately, the results do not improve by simply switching the remapping procedure. Even more important, INT2LM comprises more functionalities than only remapping the data (see answer A4 in reply to referee # 2). Most of the deviations, which prohibit the dynamical 2-way nesting at the moment, stem from the pressure adaption routine and the boundary layer treatment, not from the remapping procedure. To make this even clearer we change the sentence to "GRID can be used for all grid mapping operations required during a simulation. Nevertheless, in MMD it is currently used for the parent-to-client coupling only."

**8) p.10, l.12: change "serveral" for "several"**

Changed.

**9) p.12, l.19: please correct sentence "GRID_TRAFO comprises the in EMAC ..."**

Reformulated to "GRID_TRAFO comprises NREGRID (Jöckel, 2006), the standard remapping tool in EMAC, and the SCRIP software (Jones, 1999)."

**10) p.13, 1st parag of 3.1.2; please rephrase "It is for the rediscretisation ..."; maybe: "It is used for the rediscretisation ..."**

Added.

**11) p.19, l.6: change "aver" of "over"**

Changed.

Best regards,
Astrid Kerkweg (also on behalf of all co-authors)

| Name | Configuration | DYN | | 139 Tracer | |
|------|---------------|-----|-----|-----------|-----|
| | EMAC - CM1 - CM2 | 1-way | 2-way | 1-way | 2-way |
| S01 | 6x1 - 5x6 - 6x6 | 0.488 | 0.512 (4.9%) | 1.520 | 2.120 (39.5%) |
| S02 | 6x1 - 4x6 - 7x6 | 0.468 | 0.480 (2.6%) | 1.500 | 2.180 (45.3%) |
| S03 | 6x1 - 3x6 - 8x6 | 0.494 | 0.532 (7.7%) | 1.580 | 2.240 (41.8%) |
| S04 | 6x1 - 2x6 - 9x6 | 0.524 | 0.530 (1.1%) | 1.680 | 2.320 (38.1%) |

Table 1: Node hours required for a 1-day simulations with a MECO(2) setup employing EMAC in T42L31ECMWF, COSMO/MESSy with 120x120 horizontal grid points, 40 levels, and COSMO/MESSy 91x91 horizontal grid points, 40 levels. The "configuration" column denotes the task distribution between EMAC, the first COSMO/MESSy instance (CM1) and the second COSMO/MESSy instance (CM2). The right most 4 columns list the run time in node hours for a mere dynamical setup (DYN, 1-way vs. 2-way nested) and a setup including 139 1-way and 2-way nested tracers (but without chemical kinetics). Additionally, the column for the 2-way applications provide the additional run time required for the 2-way nesting in percent. The simulations have been performed at the "mistral" computer system at the German Climate Computing Center. The node hours are taken from the scheduler output.

**Supplementary material**

A. Kerkweg et al.

This document is available as electronic supplement of our article "The on-line coupled atmospheric chemistry model system MECO(n) - Part 5: Expanding the Multi-Model-Driver (MMD v2.0) for 2-way data exchange including data interpolation via GRID (v1.0)" in Geosci. Model Dev. (2017), available at: `http://www.geosci-model-dev.net`

**Impact of orography differences on the vertical profile**

Interpolating back and forth between differently resolved orography heights in the coupled models cause a displacement of the tracer with height. To visualise these differences, a MECO(2) simulation with a passive tracer was performed. The initial tracer distribution is horizontally homogeneous and vertically increasing. Fig. S1 displays at four different locations, the height profiles of the tracer in the parent domain (black), in the child domain (blue, circles) and the coupled field (red, upside down triangles). The annotation gives the surface height in the parent and the child domain, respectively. The blue and the black line are always on top of each other indicating the tracer is initialised with exactly the same height profile in both COSMO instances. With increasing surface height difference, the differences in the vertical profiles increase. The second row of Fig. S1 displays the differences of the black and the red line, i.e., of the original profile and the profile given by the coupling field.

[Figure]

Figure S1: Vertical profiles of a passive tracer (upper row) and its differences (lower row) for different topographic heights in the two COSMO/MESSy model instances (in $10^{-10}mol/mol$). The title indicates the topographic height in the parent / child domain, respectively. Black line: initial profile in the parent model; blue line (circles): initial profile in the child model; red line (topdown triangles): coupled tracer profile in the parent model. The lower row displays the differences between the black and the red lines. Note: the blue line is always on top of the black line.

**Example:**
**Performance of 2-way nesting for one specific setup**

The run-time performance depends very much on the chosen setup. The largest impacts are

- the complexity of the simulation itself (e.g., whether or not chemistry is included),

- the number of coupled fields,

- the sizes of the coupled domains,

- the number of chosen tasks, and

- the task distribution between the different model instances.

Thus, all numbers that might be shown here, are only valid for exactly the one specific setup as analysed in the example. We apprehend that these numbers might be misleading and could be wrongly interpreted. Therefore we are hesitating to show such numbers in the main body of the article. However, as a compromise, we include the analysis given below in the supplement, and add a reference to it in Sect. 5.

Table 1 shows the run time in node hours for a 1 day simulation of one specific MECO(2) setup, applied with four different task distributions (S01-S04) on 3 nodes (à 24 tasks) at the Mistral computer at Deutsches Klimarechenzentrum (DKRZ). For the mere dynamical setups S02 is the fastest, as could be expected from the number of horizontal grid boxes in CM1 and CM2 and the fact that only the child model has to perform the data transformations. This is still the case for the 1-way simulations including 139 tracers. However, the additional time required for the 2-way coupling is largest in the case with 139 tracers for this setup (45.3%). This shows that the required additional time for the 2-way nesting is extremly setup dependent. Thus, the best task distribution needs to be chosen by the user for each setup individually. Here, the user has to take into account at least the size of the different model domains, the number of 1-way and / or 2-way coupled variables, and the complexity of the simulation, e.g. whether or not chemistry is included.
On the first glance the required additional time seems to be large. However, the tracers in this example are only transported. No chemical kinetic calculations are included. Including chemistry, at least 10 times higher run times are to be expected. Thus the additional run time for the 2-way nesting would shrink dramatically relative to the overall run time of chemical applications.

| Name | Configuration | DYN | | 139 Tracer | |
| | EMAC - CM1 - CM2 | 1-way | 2-way | 1-way | 2-way |
|------|---------------|-------|-------|--------|-------|
| S01 | 6x1 - 5x6 - 6x6 | 0.488 | 0.512 (4.9%) | 1.520 | 2.120 (39.5%) |
| S02 | 6x1 - 4x6 - 7x6 | 0.468 | 0.480 (2.6%) | 1.500 | 2.180 (45.3%) |
| S03 | 6x1 - 3x6 - 8x6 | 0.494 | 0.532 (7.7%) | 1.580 | 2.240 (41.8%) |
| S04 | 6x1 - 2x6 - 9x6 | 0.524 | 0.530 (1.1%) | 1.680 | 2.320 (38.1%) |

Table 1: Node hours required for a 1-day simulations with a MECO(2) setup employing EMAC in T42L31ECMWF, COSMO/MESSy with 120x120 horizontal grid points, 40 levels, and COSMO/MESSy 91x91 horizontal grid points, 40 levels. The "configuration" column denotes the task distribution between EMAC, the first COSMO/MESSy instance (CM1) and the second COSMO/MESSy instance (CM2). The right most 4 columns list the run time in node hours for a mere dynamical setup (DYN, 1-way vs. 2-way nested) and a setup including 139 1-way and 2-way nested tracers (but without chemical kinetics). Additionally, the column for the 2-way applications provide the additional run time required for the 2-way nesting in percent. The simulations have been performed at the "mistral" computer system at the German Climate Computing Center. The node hours are taken from the scheduler output.

[revised manuscript text omitted]

---

## Author Response (AR3)

Dear Sophie,

thank you very much for the careful reading of our manuscript and the valuable additional comments! Please find our answers below.

**Dear author,**

**Thank you very much for your revised manuscript that answers basically all my concerns. I just have the few minor points that I think you should address before publication.**

- **Figure S1: I think it would be better to remove the sentences "Black line: initial profile in the parent model" and "Note: the blue line is always on top of the black line" and replace "blue line (circles): initial profile in the child model" by "blue line (circles): initial profile in the child model and in the parent model".**
  According to your comment changes have been made to the figure caption and the main body text.

- **p1, l17: correct "specificially"**
  Changed to "specifically".

- **p3, l3: change "patches approach" for "patch approach"**
  We changed "patches approach" to "patch approach" throughout the manuscript.

- **p5, l28: correct "tranfer"**
  This and additional "tranf" typos are corrected to "transf" throughout the manuscript.

- **p11, l23: I don't think ICON can be qualified as a structured grid, cf https://www.dkrz.de/SciVis/hd-cp-2/en-icon_grid.pdf**
  Your are right: "structured" changed to "unstructured".

- **Fig 5 & 6: Use lower case to refer to panel; for d) put (a - c)/a * 100.**
  Changed.

- **Fig 6, I don't see that data has been masked, as indicated in the captions**
  Sorry! Somehow we linked the unmasked image. Changed.

- **p19, l15: I am not sure "exemplarily" is an English word?**
  Indeed, "exemplarily" is an english word (see for example:
  http://www.oed.com/view/Entry/66042?redirectedFrom=exemplarily#eid
  or
  https://en.oxforddictionaries.com/definition/exemplarily
  As we do not find another word or phrase saying exactly the same, we'd like keep it.

Best regards,
Astrid Kerkweg (also on behalf of all co-authors)

[revised manuscript text omitted]

**Supplementary material**

A. Kerkweg et al.

This document is available as electronic supplement of our article "The on-line coupled atmospheric chemistry model system MECO(n) - Part 5: Expanding the Multi-Model-Driver (MMD v2.0) for 2-way data exchange including data interpolation via GRID (v1.0)" in Geosci. Model Dev. (2018), available at: `http://www.geosci-model-dev.net`

**Impact of orography differences on the vertical profile**

Interpolating back and forth between differently resolved orography heights in the coupled models cause a displacement of the tracer with height. To visualise these differences, a MECO(2) simulation with a passive tracer was performed. The initial tracer distribution is horizontally homogeneous and vertically increasing. Fig. S1 displays at four different locations, the height profiles of the tracer in the parent domain, in the child domain (blue, circles) and the coupled field (red, upside down triangles). The annotation gives the surface height in the parent and the child domain, respectively.  For this specific case parent model and child model initial profiles are identical, indicating that the tracer is initialised with exactly the same height profile in both COSMO instances. With increasing surface height difference, the differences in the vertical profiles increase. The second row of Fig. S1 displays the differences of the  blue and the red line, i.e., of the original profile and the profile given by the coupling field.

[Figure]

Figure S1: Vertical profiles of a passive tracer (upper row) and its differences (lower row) for different topographic heights in the two COSMO/MESSy model instances (in $10^{-10} mol/mol$). The title indicates the topographic height in the parent / child domain, respectively.  Blue line  (circles): initial profile in the child model and in the parent model; red line (topdown triangles): coupled tracer profile in the parent model. The lower row displays the differences between the  blue and the red lines.

**Example:**
**Performance of 2-way nesting for one specific setup**

The run-time performance depends very much on the chosen setup. The largest impacts are

- the complexity of the simulation itself (e.g., whether or not chemistry is included),

- the number of coupled fields,

- the sizes of the coupled domains,

- the number of chosen tasks, and

- the task distribution between the different model instances.

Thus, all numbers that might be shown here, are only valid for exactly the one specific setup as analysed in the example. We apprehend that these numbers might be misleading and could be wrongly interpreted. Therefore we are hesitating to show such numbers in the main body of the article. However, as a compromise, we include the analysis given below in the supplement, and add a reference to it in Sect. 5.

Table 1 shows the run time in node hours for a 1 day simulation of one specific MECO(2) setup, applied with four different task distributions (S01-S04) on 3 nodes (à 24 tasks) at the Mistral computer at Deutsches Klimarechenzentrum (DKRZ). For the mere dynamical setups S02 is the fastest, as could be expected from the number of horizontal grid boxes in CM1 and CM2 and the fact that only the child model has to perform the data transformations. This is still the case for the 1-way simulations including 139 tracers. However, the additional time required for the 2-way coupling is largest in the case with 139 tracers for this setup (45.3%). This shows that the required additional time for the 2-way nesting is extremly setup dependent. Thus, the best task distribution needs to be chosen by the user for each setup individually. Here, the user has to take into account at least the size of the different model domains, the number of 1-way and / or 2-way coupled variables, and the complexity of the simulation, e.g. whether or not chemistry is included.

On the first glance the required additional time seems to be large. However, the tracers in this example are only transported. No chemical kinetic calculations are included. Including chemistry, at least 10 times higher run times are to be expected. Thus the additional run time for the 2-way nesting would shrink dramatically relative to the overall run time of chemical applications.

| Name | Configuration EMAC - CM1 - CM2 | DYN | | 139 Tracer | |
|------|------|------|------|------|------|
| | | 1-way | 2-way | 1-way | 2-way |
| S01 | 6x1 - 5x6 - 6x6 | 0.488 | 0.512 (4.9%) | 1.520 | 2.120 (39.5%) |
| S02 | 6x1 - 4x6 - 7x6 | 0.468 | 0.480 (2.6%) | 1.500 | 2.180 (45.3%) |
| S03 | 6x1 - 3x6 - 8x6 | 0.494 | 0.532 (7.7%) | 1.580 | 2.240 (41.8%) |
| S04 | 6x1 - 2x6 - 9x6 | 0.524 | 0.530 (1.1%) | 1.680 | 2.320 (38.1%) |

Table 1: Node hours required for a 1-day simulations with a MECO(2) setup employing EMAC in T42L31ECMWF, COSMO/MESSy with 120x120 horizontal grid points, 40 levels, and COSMO/MESSy 91x91 horizontal grid points, 40 levels. The "configuration" column denotes the task distribution between EMAC, the first COSMO/MESSy instance (CM1) and the second COSMO/MESSy instance (CM2). The right most 4 columns list the run time in node hours for a mere dynamical setup (DYN, 1-way vs. 2-way nested) and a setup including 139 1-way and 2-way nested tracers (but without chemical kinetics). Additionally, the column for the 2-way applications provide the additional run time required for the 2-way nesting in percent. The simulations have been performed at the "mistral" computer system at the German Climate Computing Center. The node hours are taken from the scheduler output.